# Comparison of the efficacy among different interventions for radiodermatitis: A Bayesian network meta-analysis of randomized controlled trials

Ying Guan [1☉]*, Shuai Liu[2,3☉], Anchuan Li[4], Wanqin Cheng[5]

1 Department of Radiation Oncology, Guangxi Medical University Cancer Hospital, Nanning, Guangxi, P.R. China, 2 Department of Radiotherapy Oncology, The Sixth Affiliated Hospital, Sun Yat-sen University, Guangzhou, P. R. China, 3 Biomedical Innovation Center, The Sixth Affiliated Hospital, Sun Yat-sen University, Guangzhou, P. R. China, 4 Department of Radiation Oncology, Fujian Medical University Union Hospital, Fuzhou, P. R. China, 5 Department of Radiation Oncology, Shunde Hospital, Southern Medical University, Shunde, P. R. China

☉ These authors contributed equally to this work.
* 386927552@qq.com

**Data Availability Statement:** All relevant data are within the manuscript and its Supporting Information files.

## Abstract

### Background

Radiation dermatitis (RD) is a prevalent and difficult-to-manage consequence of radiation therapy (RT). A variety of interventions have been proven effective in preventing and treating RD. However, the optimal approach remains unclear. This network meta-analysis (NMA) conducted a comparison and ranking of the effectiveness and patient-reported outcomes (PROs) of the interventions currently utilized in RD.

### Methods

PubMed, Web of Science, Embase, and Cochrane Library were searched to identify pertinent randomized controlled trials (RCTs) focused on the prevention and treatment of RD. The primary outcome measures included the incidence of grade≥2 RD (i.e., percentage of moist desquamation) and RD score. The secondary outcome measures encompassed patients' subjective assessment scores of pains, itching and burning sensations.

### Results

Our meta-analysis encompassed 42 studies and 4884 participants. Regarding the primary outcomes, photobiomodulation treatment (PBMT) ranked first in surface under curve cumulative ranking area (SUCRA:0.92) for reducing the incidence of grade≥2 RD. It demonstrated a significant difference when compared to Trolamine (OR 0.18,95%CrI 0.09–0.33) and Xonrid® (OR 0.28,95%CrI 0.12–0.66). Mepitelfilm (SUCRA: 0.98) achieved the highest rank in reducing the RD score, demonstrating superiority over StrataXRT® (MD -0.89, 95% CrI -1.49, -0.29). Henna (SUCRA: 0.89) demonstrated the highest effectiveness in providing pain relief, with a significant difference compared to Hydrofilm (MD -0.44, 95% CrI -0.84,

**Funding:** This research was supported by the Youth Science Foundation of Guangxi Medical University (No. GXMUYSF201516). The funders of this work had no role in study design, data collection and analysis, decision to publish, or preparation of the manuscript.

**Competing interests:** The authors have declared that no competing interests exist.

-0.04) and Mepitelfilm (MD -0.55, 95% CrI -0.91, -0.19). Hydrofilm (SUCRA: 0.84) exhibited the fewest itching sensations, demonstrating superiority over Mepitelfilm (MD -0.50, 95% CrI -0.84, -0.17). No statistically significant difference was observed among various interventions in the assessment of burning sensations.

## Conclusion

PBMT and Mepitelfilm demonstrated better efficacy in reducing the incidence of grade$\geq$2 RD and RD score, respectively. In terms of PROs, Henna and Hydrofilm had fewer complaints in pain and itching sensations, respectively. However, studies with larger sample size on different interventions are warranted in the future.

## Trial registration

PROSPERO registration number CRD42023428598.

## Introduction

Radiation therapy (RT) is the third most frequently used cancer treatment modality. Among the side effects of RT, radiation dermatitis (RD) is a common occurrence, affecting up to 95% of patients either during or after undergoing RT [1–3]. It is characterized by varying degrees of skin inflammation and damage, ranging from mild erythema to severe ulceration and necrosis. Patients undergoing RT commonly experience grade 2 or higher RD, which manifests as notable erythema, moist desquamation, edema, and is associated with significant discomfort, pain, itching, burning, and morbidity [4]. Additionally, RD can hinder the effectiveness of RT by imposing limitations on the prescribed dose and duration of treatment. Therefore, the prevention and management of RD represent essential and significant clinical challenges in the field of radiation oncology [1, 4].

The underlying mechanisms of RD are well comprehended. Ionizing radiation induces damage to the dermis, resulting in vasodilation and the release of histamine-like substances and activating inflammatory pathways and causing cytokines overproduction; which contribute to the onset of erythema [5]. The DNA damage caused by radiation leads to a depletion of dividing basal stem cells in the epidermis. Consequently, the skin experiences thinning as the superficial layers, which are normally shed, are inadequately replaced by the deeper basal cell layer [6]. This process ultimately leads to both dry and moist desquamation. The emergence of late skin effects can be attributed to imbalances between profibrotic and proinflammatory cytokines [7].

The development of RD is influenced by various risk factors, including patient characteristics and radiation treatment-related factors. Treatment-related factors encompass parameters such as total dose, fractionation schedule, use of bolus or boost, treatment surface area or volume, radiation technique, and concurrent chemotherapy or targeted agents. Intrinsic factors like chronic sun exposure, breast size, smoking status, diabetes, and obesity may also impact the severity of RD [4, 8–11].

Multiple interventions have been proposed and demonstrated to be effective in the prevention and management of RD across various categories. These interventions include topical corticosteroids (e.g., hydrocortisone) [6], barrier films and dressings (e.g., water-based, silicone-based, and silver-based) [7,12], oral supplements (e.g., glutamine) [13], photobiomodulation

treatment (PBMT) in the form of laser therapies, light emitting diode photomodulation, and red light phototherapy [14–16], standard care measures (e.g., hygiene practices, moisturizers, and topical sprays) [17], natural agents like Chamomile, Calendula, Curcumin, and Henna [18–23], as well as miscellaneous agents such as trolamine and hyaluronic acid [24–26]. All these interventions share similar clinical actions aimed at promoting the wound healing process by restoring the physiological hydration levels of the affected skin areas and enhancing their biological activity. They possess antioxidant, anti-inflammatory, and antimicrobial properties without exerting any tumor protection effects.

However, the efficacy of these interventions for RD and their impact on patient-reported outcomes (PROs) are still subjects of debate. There is currently a lack of consensus regarding the optimal approach to RD management. Previous traditional meta-analyses have focused on evaluating the clinical effectiveness of interventions in comparison to standard care. However, these approaches have limitations as they provide only partial insights into the overall hierarchy of interventions. This is due to the fact that intervention effects are estimated and presented for only a subset of relevant intervention comparisons. Furthermore, they have reported conflicting results and limitations, such as heterogeneity, bias, and small sample sizes [7, 27–30]. Last but not least, most studies have focused on the individual categories of interventions rather than comparing them directly, which makes it difficult to determine the relative efficacy of different interventions for clinician decision-makers.

To tackle these challenges, a network meta-analysis (NMA) offers a solution by enabling a thorough and quantitative comparison of the efficacy of various interventions for RD [31–33]. As of now, no NMA has been conducted to assess the effectiveness of various interventions in the prevention and treatment of RD. Hence, the objective of this study was to employ a Bayesian framework to compare the clinical effectiveness of various interventions in the prevention and treatment of RD. The study aimed to estimate the relative efficacy of these interventions and rank them based on their probabilities of being the best, second-best, and so on [34]. The findings of this study hold significant implications for clinical practitioners in terms of the prevention and management of RD. By identifying superior interventions for RD, the study has the potential to enhance the quality of life (QOL) and treatment outcomes of patients undergoing RT. Moreover, the study may contribute to the development of evidence-based guidelines and recommendations for the prevention and management of RT.

## Methods

### Selection criteria and study design

The inclusion criteria for this study were as follows: (1) the investigation of an intervention with at least two RCTs evaluating its efficacy in the prevention and treatment of RD, and (2) the assessment of the intervention compared to a standard of care (SOC). The SOC, serving as the reference group in this context, encompasses several aspects of the following treatments: (a) Shame treatment, placebo, routine nursing methods, usual care, standard skin care, usual supportive care, normal care, or no treatment. (b) Institutional nursing staff utilized the currently recommended standard skin care treatment and standard products as per hospital protocols or institutional standards for the trials. These products include hydrocortisone, urea cream, HPRPlusTM, Biafine® cream, white Vaseline (PROPETO), Sorbolene® cream, as well as moisturizer/moisturizing base cream, aqueous creams, and other conventional products.

We excluded studies that met any of the following criteria: (1) Study design type is non-randomized controlled trial (non-RCT), including conference papers, summaries, abstracts, letters, protocols, editorials, guidelines, review articles, comments, clinical observations, case

studies, cohort studies, non-randomized trials, case-control studies, non-comparative studies, and any non-experimental investigations such as cross-sectional and retrospective surveys; (2) studies investigating an intervention with only a single RCT; (3) studies where the reference group did not correspond to the SOC; (4) duplicate studies; (5) studies for which the full text was not available; (6) studies with incomplete or unreported data; (7) studies not written in English.

The prespecified primary outcomes of the study included the incidence of grade≥2 RD (i.e., percentage of moist desquamation) and RD score. The secondary outcomes comprised subjective assessment scores of PROs such as pains, itching, and burning sensations. The outcomes of interest encompassed the RD score and patients' subjective assessment scores for overall treatment. Higher scores on the designated rating scale indicated increased skin toxicity.

All scores were defined as the highest intensity changes recorded on the rating scales presented in Table 1, at the conclusion of radiation treatments, typically occurring between five to seven weeks. The maximum level indicated the most severe skin reaction associated with a specific intervention or no intervention. These widely reported outcome measures at these particular time points hold clinical significance as the radiation dose reaches its peak cumulative level upon completion of radiation treatment. The results were recorded as closely as possible to the end of radiation therapy for all analyses. In cases where data were unavailable at the end of radiation therapy, data from time points near the conclusion of treatment were prioritized for primary outcomes, as well as secondary outcomes.

## Search strategy

A comprehensive literature search was conducted using electronic databases including PubMed, Web of Science, and Embase (the period from 1964 to May 2023). Additionally, the Cochrane Central Register of Controlled Trials (CENTRAL) in the Cochrane Library (Issue 5 of 12 in May 2023), was searched. The medical subject headings (Mesh), Emtree and entry terms were used as search strategies. The complete search strategy and the results can be found in S1 Appendix.

## Study selection and data collection

Two researchers, YG and SL independently screened the titles and abstracts of the potentially eligible studies for initial retrieval. Subsequently, both researchers thoroughly read the full texts, extracted relevant data, and engaged in discussions to reach a consensus on inclusion validity. Any disagreements were resolved through team discussions (YG, SL, ACL, and WQC). A formal extraction record form was developed to ensure systematic data extraction. The following patient and study characteristics were extracted: cancer site, average age, gender, author, year of publication, country, sample size, study design, RD grading scales, treatment arms, frequency and duration of interventions, and outcomes. A summary of the baseline characteristics of the included RCTs is provided in Table 1 [18, 35–75].

Three-arm trials that involved two different interventions were divided into the corresponding number of pairwise comparisons, comparing each intervention against the reference group (SOC).

## Quality assessment and risk of bias

Two reviewers, YG and SL, assessed the risk of bias in the selected RCTs using the Cochrane Collaboration tool within Review Manager 5.4.1. The criteria for evaluating bias included random sequence generation, allocation concealment, blinding of participants and researchers,

Table 1. Baseline characteristics of included randomized controlled trials (RCTs).

| Author | Year of publication | Country | RCTs design | Sample size (Number randomized) | Gender (M/F) | Mean±SD/ Median (Range) age (Arm1 vs Arm2 vs Arm3, years) | Cancer Site | Intervention(Frequency of application and duration) | RD grading assessment | Evaluation tools of patients' subjective assessment scores relief | Outcomes [Responders (%), Mean±SD/Mean (95%CI) /Median (Range),95%CI, maximum level] Arm1 vs Arm2 vs Arm3, n = Number analysed |
|---|---|---|---|---|---|---|---|---|---|---|---|
| Ferreira EB [18] | 2020 | Brazil | NA | Arm1(24pts): Chamomile Arm2(24pts): SOC | 33/15 | 56±16 vs 59±13 | Head and neck | Chamomile = Chamomile gel; SOC = Urea Cream, which is used in the institutional usual care; Application: apply the assigned product topically, 3 times a day, on the skin at the irradiated area, during the entire period of the RT; Duration6-8weeks. | RTOG criteria, with a range of 0–4 | participant self-report of presence of discomfort yes/no | F8: 12(n = 24) vs 17(n = 24); F9: 11(n = 24)vs 14(n = 24); |
| Ahn S [35] | 2020 | Korea | Single-centre, unblinded, parallel | Arm1(21pts): StrataXRT® Arm2(28pts): SOC | 0/49 | 46(40–56) vs 49(29–60) | Breast | StrataXRT® is a film-forming silicone gel designed to promote a moist wound-healing environment; SOC = X-derm® is a moisturizing cream (Pharmbio Korea Inc., Republic of Korea), which is an institutional usual care; Application: apply to the designated treatment site at least twice daily, starting on the first day of RT and for 4 weeks after completion of RT Duration:10weeks. | RTOG and CTCAE criteria, with a range of 0–4 | 5-point scale questionnaire with a range of 0–5 | F2: No Grade2 RD, 0(n = 21) vs 0 (n = 28); F7: sore (0–5): score4:0(n = 21) vs 0(n = 28); F8: sore (0–5): score5:0(n = 21) vs 1(n = 28); F9: sore (0–5): score5:1(n = 21) vs 0(n = 28); |
| Chan RJ [36] | 2019 | Australia | Single-blind | Arm1(100pts): StrataXRT® Arm2(97pts): SOC | 154/43 | 64.0 vs 63.6 | Head and neck | StrataXRT®; SOC = Sorbolene® cream (10% Glycerine), which is used as standard care; Application: twice a day onset of radiotherapy, onset of radiotherapy until the skin reaction subsided, up to 4 weeks post treatment; Duration: 10-11weeks. | CTCAE criteria version 4.0, with a range of 0–4 | Brief Pain Inventory (BPI), with a range of 0–10; Itching was scored on a numeric analogue scale of 0–10 | F2: 86(n = 99) vs89(n = 95); Mean(95%CI): F5: 2.4(2.2 –2.6, n = 98) vs 2.7 (2.5 –3.0, n = 91) F7: 4.066(2.913– 5.677, n = 98) vs 4.716(3.382– 6.575, n = 91); F8:2.560(1.777– 3.687, n = 98) vs 2.646(1.826– 3.835, n = 91); |
| Behroozian T [37] | 2022 | Canada | Multicenter, open-label, phase III | Arm1(251pts): Mepitelfilm Arm2(125pts): SOC | 1/376 | 58.2±11.7 vs 59.5±13.4 | Breast | Mepitelfilm = Mepitel film; SOC = Standard of care; Application: apply to the designated treatment site, starting on the first day of RT and for 2weeks after completion of RT; Duration6-7weeks. | CTCAE criteria version 5.0, with a range of 0–3 | Patient-assessed RISRAS criteria was a subject report composing four items on a 4-point scale with a possible rang 0–3. | F2: 39(n = 251) vs 57(n = 125); F7: 0.8±0.8 (n = 251) vs 1.1 ±0.8(n = 123); F8: 0.8±0.8 (n = 251) vs 1.0 ±0.8(n = 123); F9: 0.5±0.8 (n = 251) vs 0.7 ±0.8(n = 123); |

(Continued)

**Table 1.** (Continued)

| Author | Year of publication | Country | RCTs design | Sample size (Number randomized) | Gender (M/F) | Mean±SD/Median (Range) age (Arm1 vs Arm2 vs Arm3, years) | Cancer Site | Intervention(Frequency of application and duration) | RD grading assessment | Evaluation tools of patients' subjective assessment scores relief | Outcomes [Responders (%), Mean±SD/Mean (95%CI)/Median (Range),95%CI, maximum level] Arm1 vs Arm2 vs Arm3, n = Number analysed |
|---|---|---|---|---|---|---|---|---|---|---|---|
| Wooding H [38] | 2018 | New Zealand | Own control | Treatment: Arm1(22pts): Mepitelfilm Arm2(22pts): SOC Prevention: Arm1(11pts): Mepitelfilm Arm2(11pts): SOC | Treatment:17/5; Prevention:10/1 | NA | Head and neck | Treatment: Mepitelfilm = Mepitel Film; SOC = Sorbolene cream, which is declared institutional preference; Prevention: Mepitelfilm = Mepitel Film; SOC = Biafine® cream, which is declared institutional preference; Application: Prevention: started using Film and Cream from commencement of radiation therapy. Treatment: started using Film and Cream from the moment faint erythema was visible. Film was applied by the researcher on the skin area randomized to film; Cream was applied twice daily to the control area by the patients; Duration: 5-7weeks. | RTOG criteria, with a range of 0–3 | NA | F2: Treatment:20 (n = 22) vs 21 (n = 22); Prevention:7 (n = 11) vs 10 (n = 11); |
| Zhong WH [39] | 2013 | China | NA | Arm1(43pts): Mepitelfilm Arm2(45pts): SOC | 55/33 | NA | Nasopharyngeal carcinoma (NPC) | Mepitelfilm = Mepilex Lite; SOC = Usual care; Application: the healing of post irradiation dermatitis in NPC patients; Duration: until wound was healed. | NA | Wound pain was assessed with a 0–10 cm visual analog scale (VAS), with a range of 0–10 | F7: 5.31±1.26 (n = 43) vs 6.89 ±1.87(n = 45); |
| Moller PK [40] | 2018 | Denmark | Own control | 79pts: intra-patient control Arm1: Mepitelfilm Arm2:SOC | 0/79 | 61.9 | Breast | Mepitelfilm = Mepitel Film; SOC = Standard care; Application: 1st time of RT to end of RT; Duration: 3weeks. | RTOG/EORTC scale with a range of 0–4 | patient-reported outcome measures (PROM) regarding skin symptoms, used a 4-Point Likert Scale | F2: 5(7%), n = 76 vs 11(14%), n = 76; F7: 0(n = 78) vs 4 (n = 78; F8:2(n = 77) vs 5 (n = 77); F9: 1(n = 78) vs 3 (n = 78); |
| Rades D [41] | 2019 | Germany | NA | Arm1(28pts): Mepitelfilm Arm2(29pts): SOC | 46/11 | NA | Head and neck | Mepitelfilm = Mepitel Film (MEP); SOC = Standard skin care; Application: changed twice per week; Day1 of RT until 1w after treatment or until grade≥2 moist desquamation or grade 3 radiation dermatitis occurred; Duration: 9weeks. | CTCAE criteria version 4.03, with a range of 0–4 | A self-rating scale developed by authors, with a range of 0–10 | F2: 4(57.1%), n = 7 vs 15 (57.7%), n = 26; F7: 2.0(0–6), n = 7 vs 2.5(0–8), n = 26; |
| Yan J [42] | 2020 | China | Own control | 44pts: intra-patient control Arm1: Mepitelfilm Arm2:SOC | 28/11(number analysed) | 54 | Head and neck | Mepitelfilm = Mepitel Film; SOC = Biafine® cream, which is used at Drum Tower Hospital as standard of care; Application: Mepitelfilm: applied to the side of the neck randomized to Mepitel Film, replaced if it came off the skin overnight or if significant areas curled up at the edges. SOC: twice daily to the control side of the neck; Duration: 5weeks. | F2: RTOG criteria, with a range of 0–3; F5: RISRAS criteria with a total rang 0–36 | NA | F2: 21(54%), n = 39 vs 33 (85%), n = 39; F5: 2.83±0.16, n = 39vs 4.02 ±0.20, n = 39; |

(Continued)

**Table 1.** (Continued)

| Author | Year of publication | Country | RCTs design | Sample size (Number randomized) | Gender (M/F) | Mean±SD/ Median (Range) age (Arm1 vs Arm2 vs Arm3, years) | Cancer Site | Intervention(Frequency of application and duration) | RD grading assessment | Evaluation tools of patients' subjective assessment scores relief | Outcomes [Responders (%), Mean±SD/Mean (95%CI) /Median (Range),95%CI, maximum level] Arm1 vs Arm2 vs Arm3, n = Number analysed |
|---|---|---|---|---|---|---|---|---|---|---|---|
| Abbas H [43] | 2012 | Egypt | Phase III | Arm1(15pts): Trolamine Arm2(15pts): SOC | 25/5 | Median:53.2 vs 55.8 | Head and neck | Trolamine = Trolamine emulsion; SOC = Usual supportive care; Application: apply trolamine 3 times daily beginning on the first day of RT, continuing throughout RT, and for 2 weeks after RT completion. Trolamine was applied at 8-h intervals and 4 hours before and after radiation; Duration:8–9weeks. | RTOG criteria, with a range of 0–4 | NA | F2: 3(n = 15) vs 8 (n = 15); |
| Elliott EA [44] | 2006 | Canada | Three-arm | Arm1(166pts): Trolamine (Prevention) Arm2(175pts): Trolamine (Treatment) Arm3(165pts): SOC | 400/106 | 59.1 vs 59.5 vs 58.8 | Head and neck | Trolamine = Medix Pharmaceuticals (Americas Inc, Largo, FL) supplied; SOC = Institutional Preference/ Standard of care; Application: Prevention: three times daily on the 1st day of RT until the end of RT; Treatment: only applied once skin became itchy, bothersome, or reddened, until the end of RT; Duration:7–9weeks | NCI-CTC version 2.0 with a range of 0–4 | Head and Neck Radiotherapy Questionnaire (HNRQ), Maximum score for skin score is 18 | F2: 79%(n = 163) vs 77%(n = 172) vs 79%(n = 165); F5: [Mean = 4.8, Median = 3(0– 18), n = 143 vs [Mean = 5.3, Median = 4(0– 18), n = 157] vs [Mean = 5.7, Median = 5(0– 18), n = 130] |
| Pinnix C [45] | 2012 | United States | Single-blind, own control | 74pts: intra-patient control Arm1: Hyaluronicacid Arm2: SOC | 0/74 | 55.4 | Breast | Hyaluronicacid = Hyaluronic acid; SOC = Standard of Care; Application: three times a day, on the 1st day of RT to the end of RT; Duration: 5–8weeks. | CTCAE criteria version 3.0, with a range of 0–4 | NA | F2: 40(n = 65) vs 31(n = 65); |
| Rahimi A [46] | 2020 | United States | Double-blind, own control | 28pts: intra-patient control Arm1: Hyaluronicacid Arm2: SOC | 0/28 | Median: 60 | Breast | Hyaluronicacid = HA formulation; SOC = Control cream (placebo); Application: three times daily throughout the course of RT (Day1 of RT throughout the course of RT, discontinue use if ≥Grade 3 RD; Duration:5–6weeks. | CTCAE criteria version 4.0, with a range of 0–4 | Physician's evaluation of acute skin toxicity, examination and question | F2: 6(21%), n = 28 vs 7(25%), n = 28; F7: 3(11%), n = 28 vs 5(18%), n = 28; F8: 1(4%), n = 28 vs 4(14%), n = 28; F9: 1(4%), n = 28 vs 5(18%), n = 28; |
| Censabella S [47] | 2016 | Belgium | NA | Arm1(38pts): PBMT Arm2(41pts): SOC | 0/79 | 55.53±8.69 vs 54.54±10.01 | Breast | PBMT = MLS® laser therapy (LT); SOC = Standard of care; Application: received six sessions of LT (two times/week), starting from fraction 20 of radiotherapy; Duration:3–4weeks. | modified version of the RTOG criteria by the College of radiographers with a possible range of 0–3 | NA | F2: 1(n = 38) vs 12(n = 41); |
| Robijns J [48] | 2022 | Blegium | NA | Arm1(39pts): PBMT Arm2(32pts): SOC | 0/71 | 57.03±10.3 vs 58.72±9.6 | Breast | PBMT = Photobiomodulation; SOC = Standard skincare; Application: applied from the first until the last day of RT (2 sessions/week, 8 sessions in total); Duration:4weeks. | the modified version of the RTOG criteria, with a possible range of 0–3 | NA | F2: 4 (n = 39) vs 9 (n = 32); |

*(Continued)*

**Table 1.** (Continued)

| Author | Year of publication | Country | RCTs design | Sample size (Number randomized) | Gender (M/F) | Mean±SD/ Median (Range) age (Arm1 vs Arm2 vs Arm3, years) | Cancer Site | Intervention(Frequency of application and duration) | RD grading assessment | Evaluation tools of patients' subjective assessment scores relief | Outcomes [Responders (%), Mean±SD/Mean (95%CI) /Median (Range),95%CI, maximum level] Arm1 vs Arm2 vs Arm3, n = Number analysed |
|---|---|---|---|---|---|---|---|---|---|---|---|
| Fife D [49] | 2010 | United States | Double-blind | Arm1(18pts): PBMT Arm2(15pts): SOC | 0/33 | NA | Breast | PBMT = Light emitting diode (LED) photomodulation; SOC = Sham treatment; Application: before and after each irradiation during RT;7 additional daily treatment over next 2weeks after RT; Duration: 8-9weeks. | NCI 5-point scale for grading skin reactions with a range of 0–4 | NA | F2: 12(66.6%), n = 18 vs 9(60%), n = 15; |
| Robijns J [50] | 2018 | Belgium | NA | Arm1(60pts): PBMT Arm2(60pts): SOC | 0/120 | 56.52 vs 56.92 | Breast | PBMT = Photobiomodulation; SOC = Placebo-sham treatment; Application: 14 sessions of PBMT, 2 days a week, immediately after RT session, starting at the first day of RT; Duration:7weeks. | RTOG/EORTC scale with a range of 0–4 | NA | F2: 4(6.7%), n = 60 vs 18 (30%), n = 60; |
| Robijns J [51] | 2019 | Belgium | NA | Arm1(60pts): PBMT Arm2(60pts): SOC | 0/120 | 56.52 vs 56.92 | Breast | PBMT = Photobiomodulation; SOC = Placebo-sham treatment; Application: from the first until the last day of RT (2×/week, 14 sessions); Duration:7weeks. | RTOG/EORTC scale with a range of 0–4 | NA | F2: 4(6.7%), n = 60 vs 18 (30%), n = 60; |
| Robijns J [52] | 2021 | Belgium | NA | Arm1(28pts): PBMT Arm2(18pts): SOC | 39/7 | 64.06 vs 65.06 | Head and neck | PBMT = Photobiomodulation; SOC = Institutional skincare; Application: from the first day of RT (2/ week) until the last day of RT (14 sessions in total); Duration:7weeks. | CTCAE criteria version 4.03, with a range of 0–4 | NA | F2: 8(28.6%), n = 28 vs14 (77.8%), n = 18; |
| Zhang X [53] | 2018 | China | NA | Arm1(30pts): PBMT Arm2(30pts): SOC | 42/18 | 46.4 vs 45.23 | Head and neck | PBMT = Red light phototherapy (RLPT); SOC = Routine methods of nursing; Application: during RT, photon therapy apparatus RLPT was added, 10 min/ time, 2 times/day, and lasted until the end of RT. Duration:7-8weeks. | RTOG criteria, with a range of 0–4 | A numerical rating scales (NRS), with a range of 0–10 | F2: 12, n = 30 vs 28, n = 30; F7: 0, n = 30 vs 0, n = 30; |
| Garbuio DC [54] | 2022 | Brazil | Double-blind | Arm1(27pts): Chamomile Arm2(27pts): SOC | 0/54 | 57.41 vs 55.04 | Breast | Chamomile = Chitosan-coated chamomile microparticles formulation; SOC = Control; Application: Once a day after each session on the 1st day of RT to the end of RT; Duration: 3-5weeks. | RTOG/EORTC scale with a range of 0–4 | A visual analog scale to determine pain, itching, and burning symptoms, with a possible range of 0–10 | F2:9(33.33%), n = 27 vs 7(25%), n = 27; F7: worsening of pain score>3: 6 (n = 27) vs 14 (n = 27), $P = 0.024$; F8: worsening of itching score:4 (n = 27) vs 13 (n = 27), $P = 0.008$; F9: burning score≥8:0(n = 27) vs 3 (n = 27), $P = 0.075$ |

*(Continued)*

Table 1. (Continued)

| Author | Year of publication | Country | RCTs design | Sample size (Number randomized) | Gender (M/F) | Mean±SD/Median (Range) age (Arm1 vs Arm2 vs Arm3, years) | Cancer Site | Intervention(Frequency of application and duration) | RD grading assessment | Evaluation tools of patients' subjective assessment scores relief | Outcomes [Responders (%), Mean±SD/Mean (95%CI)/Median (Range),95%CI, maximum level] Arm1 vs Arm2 vs Arm3, n = Number analysed |
|---|---|---|---|---|---|---|---|---|---|---|---|
| Schmeel LC [55] | 2019 | Germany | Own control | 74pts: intra-patient control Arm1: Hydrofilm Arm2: SOC | 0/74 | 62(37–84) | Breast | Hydrofilm = Hydrofilm (Paul Hartmann AG, Heidenheim, Germany); SOC = Urea lotion (Eucerin Urea Repair PLUS Lotion 5% Urea, Beiersdorf AG, Hamburg, Germany), which is used in the institutional usual care; Application: 1st day of RT until the completion of RT course; Duration: 4-5weeks. | CTCAE criteria version 4.03, with a range of 0–4 | Patient-assessed RISRAS criteria was a subject report composing four items on a scale of used a 4-Point Likert Scale, with a possible rang 0–3. | F2: 7(n = 74) vs 27(n = 74); F5: 0.54±0.56 (n = 74) vs 1.34 ±0.63 (n = 74); F7: mean:0.18 (n = 74) vs 0.72 (n = 74); P < 0.001 F8: mean: 0.22 (n = 74) vs 0.95 (n = 74); P < 0.001 F9: mean: 0.12 (n = 74) vs 0.74 (n = 74); P < 0.001 |
| Schmeel LC [56] | 2018 | Germany | Own control | 62pts: intra-patient control Arm1: Hydrofilm Arm2: SOC | 0/62 | Median: 62 | Breast | Hydrofilm = Polyurethane film dressings; SOC = 5% urea lotion, which is institutional usual care; Application: Hydrofilm: applied immediately preceding the 1st RT, and replaced upon detachment at least every 2weeks; SOC: twice a day starting the 1st day of RT; Duration: 5-7weeks. | RTOG/EORTC scale with a range of 0–4 | patient-assessed modified RISRAS scale, with a range of 0–3 for each item | F2: 7(12.5%), n = 56 vs 23 (41.1%), n = 56; F7: Mean pain:0.44, n = 56 vs 0.83, n = 56; P = 0.04; F8:Mean itching:0.32, n = 56 vs 1.0, n = 56;P < 0.001; F9: Mean burning:0.44, n = 56 vs 0.8, n = 56; P = 0.08; |
| Sharp L [57] | 2013 | Sweden | Blinded | Arm1(203pts): Calendula Arm2 (208pts): SOC | 0/411 | 58±11.1 vs 58 ±10.8 | Breast | Calendula = Calendula cream; SOC = Essex® cream is an aqueous cream, currently the recommended standard skin care treatment at the RT unit, Department of Oncology; Application: apply a thin layer of the assigned cream twice a day, starting at the onset of RT and continuing until two weeks after final RT session, or until the RD was healed; Duration:7-8weeks. | RTOG criteria, with a range of 0–4 | NA | F2: 45(n = 203) vs 38(n = 208); |
| Schneider F [58] | 2015 | Brazil | Double-blind | Arm1(24pts): Calendula Arm2(27pts): SOC | NA (most male) | 62.38 vs 60.44 | Head and neck | Calendula = Calendula SOC = Control group (EFA), which is the standard product for the nursing staff of the institution of this research for the prevention and treatment of these skin lesions; Application: every 12 hours (twice/day), from the 1st to the last day of RT session; Duration: 7 weeks. | RTOG criteria, with a range of 0–4 | NA | F2: 6(46.16%), n = 13 vs 3 (21.43%), n = 14; |

*(Continued)*

**Table 1.** (Continued)

| Author | Year of publication | Country | RCTs design | Sample size (Number randomized) | Gender (M/F) | Mean±SD/ Median (Range) age (Arm1 vs Arm2 vs Arm3, years) | Cancer Site | Intervention(Frequency of application and duration) | RD grading assessment | Evaluation tools of patients' subjective assessment scores relief | Outcomes [Responders (%), Mean±SD/Median (95%CI) /Median (Range),95%CI, maximum level] Arm1 vs Arm2 vs Arm3, n = Number analysed |
|---|---|---|---|---|---|---|---|---|---|---|---|
| Talakesh T [59] | 2022 | Iran | Triple-blind | Arm1(21pts): Curcumin Arm2 (21pts):SOC | 0/42 | NA | Breast | Curcumin = nanocurcumin capsules SOC = Placebo containing soybean oil; Application: 80 mg per day nanocurcumin capsules (twice a day). Each group started treatment from the first session of radiotherapy and continued two weeks after the last treatment session; Duration: 5-7weeks. | RTOG criteria, with a range of 0–4 | NA | F2: 9(n = 21) vs 18(n = 21); |
| NCT01042938 [60] | 2010 | United States | Quadruple-blind | Arm1(17pts): Curcumin Arm2(18pts): SOC | 0/35 | 50.4 vs 53.7 | Breast | Curcumin = Curcumin C3 Complex; SOC = Placebo; Application: three times daily by mouth for prescribed course of radiation treatment; Duration:4–7 weeks. | F2: Percentage of moist desquamation; F5: Radiation Dermatitis Severity (RDS) Scale, with a range of 0–4 | McGill Pain Questionnaire-Short Form (MPQ-SF) total pain score, with a range 0–50 | F2: 4(28.6%), n = 14 vs 14 (87.5%), n = 16; F5: 2.6±0.994 (n = 14) vs 3.4 ±0.554(n = 16); F7: 5.71±4.27 (n = 14) vs 3.50 ±3.43(n = 16); |
| NCT01246973 [61] | 2010 | United States | Triple-blind | Arm1(344pts): Curcumin Arm2(342pts): SOC | 0/686 | 57.6 vs 57.7 | Breast | Curcumin = Curcumin; SOC = Placebo; Application: 3 times/day throughout course of radiation treatments plus one week; Duration: 6 weeks. | F2: Percentage of moist desquamation; F5: Radiation Dermatitis Severity (RDS) Scale, with a range of 0–4 | NA | F2: 27(9.541%), n = 283 vs 36 (12.203%), n = 295; F5: 2.02±0.05, (n = 283) vs 1.99 ±0.06, (n = 295); |
| Palatty PL [62] | 2014 | India | NA | Arm1(25pts): Curcumin Arm2(25pts): SOC | 37/13 | 54.3 vs 56.9 | Head and neck | Curcumin = Vicco® turmeric cream (VTC); SOC = Johnson's®baby oil (JBO) has been regularly used anecdotally in radiation treatment in India; Application: five times a day (2 h before, immediately after, 2 h after and 4 and 6 h after radiotherapy), Day1 of RT until 2weeks after treatment; Duration: 7weeks. | RTOG/EORTC scale with a range of 0–4 | NA | F2: 3(13.63%), n = 22 vs 7 (29.2%), n = 24; |
| Ryan Wolf J [63] | 2020 | United States | Three-arm, semi-blind | Arm1(64pts): Curcumin Arm2(65pts): HPRPlus™ Arm3(62pts): Placebo arm-SOC | 0/191 | 59 vs 60.7 vs 59.8 | Breast | Curcumin = Tropic curcumin gel; HPR Plus™ which is an FDA-approved and recommended care; SOC = Placebo; Application: three times daily starting the first day of RT until 1 week after RT completion; Duration: 6-7weeks. | F2: CTCAE, with a range of 0–4; F5: Radiation Dermatitis Severity (RDS) Scale, with a range of 0–4 | Skin-Pain Inventory on a 6-point severity scale with a range of 0–5 | F2: 15(25.42%), n = 59 vs 12 (20.34%), n = 59 vs 12(22.64%), n = 53; F5: Mean (95% CI):2.68 (2.49,2.86), n = 59 vs 2.64(2.45,2.82), n = 59 vs 2.63 (2.44,2.83), n = 53; F7: Mean (95% CI):1.47 (1.12,1.81), n = 59 vs 1.44(1.10,1.78), n = 59 vs 1.64 (1.28,2.00), n = 53; |

*(Continued)*

**Table 1.** (Continued)

| Author | Year of publication | Country | RCTs design | Sample size (Number randomized) | Gender (M/F) | Mean±SD/ Median (Range) age (Arm1 vs Arm2 vs Arm3, years) | Cancer Site | Intervention(Frequency of application and duration) | RD grading assessment | Evaluation tools of patients' subjective assessment scores relief | Outcomes [Responders (%), Mean±SD/Mean (95%CI) /Median (Range),95%CI, maximum level] Arm1 vs Arm2 vs Arm3, n = Number analysed |
|---|---|---|---|---|---|---|---|---|---|---|---|
| Huang CJ [64] | 2019 | China | Double-blind | Arm1(33pts): Glutamine Arm2(31pts): SOC | 60/4 | 52.2 vs 52.6 | Head and neck | Glutamine = Oral glutamine (5g glutamine and 10 g maltodextrin) SOC = Placebo (15g maltodextrin); Application:3 times daily, 7 days before RT to 14 days after RT; Duration: 7–9weeks. | CTCAE criteria version 4.03, with a range of 0–4 | NA | F5: 1.5±0.6 (n = 33) vs 1.7 ±0.6(n = 31) |
| Lopez-Vaquero D [65] | 2017 | Spain | Double-blind | Arm1(25pts): Glutamine Arm2(25pts): SOC | 38/12 | Median: 59.0 vs 61.5 | Head and neck | Glutamine = Oral 10g glutamine; SOC = Placebo (10g maltodextrin); Application: three times daily (during RT period); Duration: 6–7weeks. | CTCAE criteria version 3.0, with a range of 0–3 | A visual analog scale (VAS), with a range of 0–10 | F2: 6(24%), n = 25 vs 13 (54.2%), n = 24; F7: 2.32, n = 25 vs 1.96, n = 24, P = 0.573 |
| Ingargiola R [66] | 2020 | Italy | NA | Breast: Arm1(20pts): Xonrid® Arm2(20pts): SOC Head and neck: Arm1(20pts): Xonrid® Arm2(20pts): SOC | Breast:0/40; Head and neck:32/8 | Breast:49.56 vs 49.01; Head and neck:55.73 vs 57.49 | Breast+ Head and neck | Xonrid® = Xonrid®; SOC = Standard of care; Application: three times daily, the first day of RT and until 2 weeks after RT completion or until the development of grade ≥ 3 skin toxicity; Duration: 8–9weeks. | CTCAE criteria version 4.0, with a range of 0–4 | NA | F2: Breast: 8(44.4%), n = 18 vs 13 (72.2%), n = 18; Head and neck 7 (35%), n = 20 vs 5(31.3%), n = 16; |
| NCT03255980 [67] | 2017 | Italy | NA | Breast: Arm1(20pts): Xonrid® Arm2(20pts): SOC Head and neck: Arm1(20pts): Xonrid® Arm2(20pts): SOC | Breast:0/40; Head and neck:32/8 | Breast:49.56 vs 49.01; Head and neck:55.73 vs 59.49 | Breast+ Head and neck | Xonrid® = Xonrid®; SOC = Standard of care; Application: three times daily, the first day of RT and until 2 weeks after RT completion or until the development of grade ≥ 3 skin toxicity; Duration: 5–7weeks. | CTCAE criteria version 4.0, with a range of 0–4 | NA | F2: Breast: 8 (n = 18) vs 13 (n = 18); Head and neck:7 (n = 20) vs 5 (n = 16); |
| Yokota T [68] | 2021 | Japan | Double-blind | Arm1(101pts): Difluprednate Arm2(102pts): SOC | 170/33 | Median: 66 vs 64 | Head and neck | Difluprednate = MYSER ointment 0.05%; SOC = Placebo group, white Vaseline (PROPETO) was selected; Application: applied at least once a day, Grade1 RD/ 30Gy of RT until 2weeks after treatment; Duration: 6–7weeks. | CTCAE criteria version 4.0, with a range of 0–4 | NCI Patient-Reported Outcomes (PRO) version of CTCAE | F2: 74(73.3%), n = 101 vs 82 (80.4%), n = 102; F8:8(8%), n = 101 vs 13(13%), n = 102; |
| Ho AY [69] | 2018 | USA | Double-blinded | Arm1(64pts): Mometasone Arm2(60pts): SOC | 0/124 | 49(26–76) vs 47.5(30–80) | Breast | Mometasone = 0.1% mometasone furoate; SOC = Eucerin Original® (E) cream, which is used in the institutional usual care; Application: applied twice daily from Day 1 of RT to 14 days post-RT; Duration:7–8weeks. | CTCAE criteria version 4.03, with a range of 0–4 | NA | F2: 28(n = 64) vs 40(n = 60); |

*(Continued)*

**Table 1.** (Continued)

| Author | Year of publication | Country | RCTs design | Sample size (Number randomized) | Gender (M/F) | Mean±SD/Median (Range) age (Arm1 vs Arm2 vs Arm3, years) | Cancer Site | Intervention(Frequency of application and duration) | RD grading assessment | Evaluation tools of patients' subjective assessment scores relief | Outcomes [Responders(%), Mean±SD/Mean (95%CI) /Median (Range),95%CI, maximum level] Arm1 vs Arm2 vs Arm3, n = Number analysed |
|---|---|---|---|---|---|---|---|---|---|---|---|
| Kiainia M [70] | 2021 | Iran | Three-arm, double-blinded | Arm1 (38pts); Mometasone Arm2(31pts): Hydrocortisone Arm3(36pts): Control arm-SOC | 0/105 | Mean (Range):47.98 (29–66) vs 55.87(36–81) vs 48.06(28–80) | Breast | Mometasone = Mometasone; Hydrocortisone: well-known care in RT department; SOC = Moisturizing base cream, which used as a control cream; Application: apply a thin layer of the cream on a daily basis, to the irradiated area from the first day of RT until week 5 Duration:5weeks. | CTCAE criteria version 4.03, with a range of 0–4 | NA | F2:18(n = 38) vs 16(n = 31) vs 19 (n = 36); |
| Miller RC [71] | 2011 | USA | Double-blinded | Arm1(85pts): Mometasone Arm2(84pts): SOC | 0/169 | 60(35–89) vs 57(27–85) | Breast | Mometasone = mometasone furoate (0.1% MMF) SOC = Placebo cream; Application: apply 3 mL of MMF cream or placebo cream lightly once daily to the area under treatment at no less than 4 hours before or after RT until completion of RT; Duration:5-6weeks. | CTCAE criteria version 3.0, with a range of 0–4 | Skindex-16, using an analogue scale (0 = never bothered to 6 = always bothered) | F2: 30(n = 84) vs 37(n = 82); F5: 1.2±0.85 (n = 84) vs 1.3 ±0.80(n = 82); F7: 1.0±1.36 (n = 83) vs 1.4 ±1.66(n = 84); F8: 1.5±1.53 (n = 83) vs 2.2 ±1.47(n = 84); F9: 1.5±1.69 (n = 83) vs 2.1 ±1.74(n = 84); |
| Sunku R [72] | 2021 | India | NA | Arm1(52pts): Betamethasone Arm2(54pts): SOC | 92/14 | NA | Head and neck | Betamethasone = betamethasone 0.1%; SOC = no treatment; Application: treated with topical betamethasone 0.1% twice daily during radiotherapy/chemo-radiotherapy; Durations:7-8weeks. | RTOG criteria, with a range of 0–4 | NA | F2: 35(n = 44) vs 41(n = 41); |
| NCT03374995 [73] | 2017 | United States | NA | Arm1 (13pts): KeraStat Arm2(11pts): SOC | 0/24 | 59.5 vs 65.36 | Breast | KeraStat = Topical keratin (KeraStat® Cream); SOC = Standard of care; Application: at least BID until the end of radiation therapy; Duration: 3–6 weeks. | F2: RTOG criteria, with a range of 0–4; F5: using RTOG Toxicity scale Grade 1 and Grade 2 assessed only, score ranges 0–2 | NA | F2: 5(38.5%), n = 13 vs 6 (54.5%), n = 11; F5: 1.00±0.0001 (n = 13) vs 1.36 ±0.64(n = 11); |
| NCT03559218 [74] | 2018 | United States | NA | Arm1(13pts): KeraStat Arm2(11pts): SOC | 0/24 | 59.2 vs 65.4 | Breast | KeraStat = Topical keratin (KeraStat® Cream); SOC = Standard of care; Application: at least BID until the end of radiation therapy; Duration: 8–12 weeks. | RTOG criteria, with a range of 0–4 | NA | F5: Mean (Range):1.00 (0–1), n = 12 vs 1.36 (0–2), n = 11 |

*(Continued)*

**Table 1.** (Continued)

| Author | Year of publication | Country | RCTs design | Sample size (Number randomized) | Gender (M/F) | Mean±SD/ Median (Range) age (Arm1 vs Arm2 vs Arm3, years) | Cancer Site | Intervention(Frequency of application and duration) | RD grading assessment | Evaluation tools of patients' subjective assessment scores relief | Outcomes [Responders (%), Mean±SD/Median (95%CI) /Median maximum level] Arm1 vs Arm2 vs Arm3, n = Number analysed |
|---|---|---|---|---|---|---|---|---|---|---|---|
| Ansari M [75] | 2013 | Iran | Phase II | Arm1(30pts); Henna Arm2(30pts); SOC | 0/60 | Median: 49.03 vs 47.03 | Breast | Henna = Alpha ointment, is extracted from natural Henna; SOC = Hydrocortisone cream (1%)), which is a well-known usual care; Application: twice a day, beginning on the day of the last session of RT; Duration: 3weeks. | CTCAE criteria version 4.0, with a range of 0–4 | Subjective scoring system developed by authors, with a range of 0–3 | F7: 1.27±0.450 (n = 30) vs 2.20 ±0.761(n = 30); F8: 1.43±0.568 (n = 30) vs 1.87 ±0.730(n = 30); F9: 1.53±0.571 (n = 30) vs 1.99 ±0.694(n = 30); |

pts = patients

CI = Confidence interval

RD = Radiodermatitis

RT = Radiotherapy

NA = Not available

F2 = Incidence of ≥Grade2 RD

F5 = RD score

F7 = Pain sensation score

F8 = Itching sensation score

F9 = Burning sensation score

RTOG = Radiation Therapy Oncology Group

CTCAE = Common Terminology Criteria-Adverse Events

NCI = National Cancer Institute

EORTC = European Organization for Research and Treatment Cancer

CTC = Common Terminology Criteria

Mepitelfilm = Mepitel film, Mepilex Lite

Trolamine = Trolamine, Trolamine cream

Hyaluronicacid = Hyaluronic acid, HA formulation

PBMT = MLS® laser therapy (LT), Light emitting diode (LED) photomodulation, Photobiomodulation therapy, Red light phototherapy (RLPT)

Chamomile = Chamomile gel

Calendula = Calendula cream

Curcumin = Nanocurcumin, Curcumin C3 complex, Curcumin cream, Vicco® turmeric cream (VTC)

Henna = Alpha ointment

Glutamine = Oral glutamine, L- Glutamine

Corticosteroid = Difluprednate (MYSER ointment 0.05%), Mometasone, 0.1% mometasone furoate, Betamethasone0.1%

KeraStat = KeraStat(R) Cream, Keratin

SOC = Standard of care, including: a) Standard skin care, standard of care, placebo, control, usual care, sham treatment, institutional preference or no intervention; b) Institutional nursing staffs used the currently recommended standard skin care treatment and standard products for prevention and treatment of these skin lesions as a part of hospital protocols or institutional standard for the trials such as hydrocortisone, urea cream, HPRPlus™, Biafine® cream, white Vaseline (PROPETO), Eucerin Original® (E) cream and Sorbolene® cream, as well as moisturizer/ moisturizing base cream, aqueous creams, and other conventional products or normal care.

blinding of outcome assessors, the integrity of outcome indicators, selective reporting, and other potential sources of bias. Each criterion was classified as having a low, unclear, or high risk of bias. A funnel plot was generated using STATA 15.1 to examine the possibility of publication bias [76]. The funnel plot revealed an asymmetric distribution, indicating the presence of publication bias. Additionally, we categorized studies with more than 100 patients per study group as large studies [77].

## Statistical analysis

Initially, the collected data were consolidated before being incorporated into the analysis. For continuous outcomes indicated by mean differences (MDs), and dichotomous outcomes indicated by odds ratios (ORs), we utilized both pairwise and network meta-analysis techniques to estimate the effect sizes. The corresponding 95% credible intervals (CrI) were also calculated. In instances where the extracted data from the studies provided information on continuous variables in the form of medians, interquartile ranges, or ranges instead of means and standard deviations, we employed the method outlined by Hozo et al. [78] to transform the data appropriately.

We performed NMA using a Bayesian framework with random effects. This analysis was conducted in the R software environment version 4.2.1, utilizing the "gemtc" package [79]. Additionally, we utilized Stata 15.1, a software package by StataCorp LLC, to generate a network plot. In the network plot, each node represents either an intervention or SOC. The size of the nodes corresponds to the number of patients, while the thickness of the lines indicates the number of trials comparing the intervention against the SOC. Moreover, we calculated the surface under the curve cumulative ranking area (SUCRA) to determine the rank probabilities of different interventions. Higher SUCRA scores indicate better efficacy for each outcome. Our protocol follows the Preferred Reporting Items for Systematic Reviews and Meta-Analyses for Protocols (PRISMA-P) guideline (S1 Checklist). It is important to note that the current study protocol was registered in PROSPERO with the registration number CRD42023428598.

## Results

### Study characteristics and quality

Out of the 1725 titles that were examined, 1077 records had their abstracts evaluated. From this group, 158 complete studies were assessed to determine their eligibility. Eventually, a total of 42 publications, which involved 4884 patients and were conducted between 2006 and 2022, met the criteria for inclusion in this analysis. We obtained the 1725 titles and abstracts from various sources such as PubMed, Web of Science, Embase, and Cochrane Library. After removing 648 articles that were duplicates, a quick review helped eliminate 919 irrelevant articles. We then proceeded to evaluate the eligibility of 158 full articles. The final selection consisted of 42 studies for qualitative synthesis, and 41 studies for quantitative synthesis (excluding Ferreira, E.B. 2020 [18], as the data provided for patients' subjective assessment scores could not be extracted or calculated as mean or standard deviation). Fig 1 illustrates the flowchart, and Table 1 provides an overview of the basic characteristics of the forty-two RCTs included.

A total of 14 different interventions were compared to SOC, namely Corticosteroid, PBMT, Glutamine, StrataXRT®, Mepitelfilm, Hydrofilm, Xonrid®, KeraStat, Chamomile, Calendula, Curcumin, Henna, Trolamine, Hyaluronicacid. These interventions represent the following classes of therapies: topical corticosteroids, PBMT, oral agents, barrier films and dressings, natural and traditional herbs, and miscellaneous agents (S1 File).

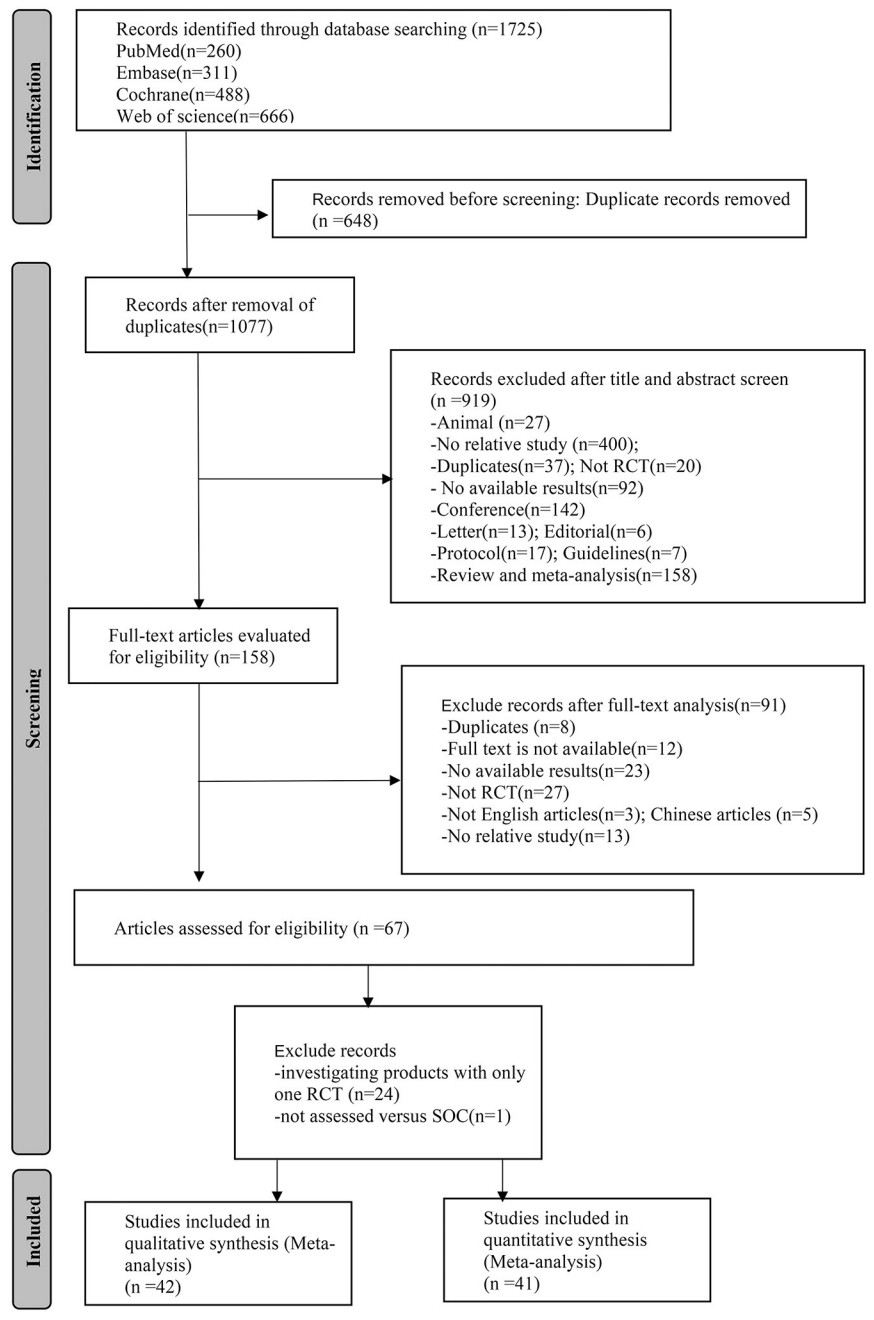

**Fig 1. Flow diagram of the study selection process.**

## Risk of bias in included studies and publication bias

Ninety-five percent (40 out of 42) of the trials included in the analysis were double-arm studies, while the remaining 5 percent (2 out of 42) were three-arm studies. The risk of bias for the included RCTs is depicted in Fig 2. We observed a low risk of bias in all of the included RCTs regarding random sequence generation, incomplete outcome data, selective reporting, and other bias. However, there was an unclear bias risk for allocation concealment in 31 RCTs. Regarding participant and staff blinding, as well as outcome assessment blinding, 20 RCTs

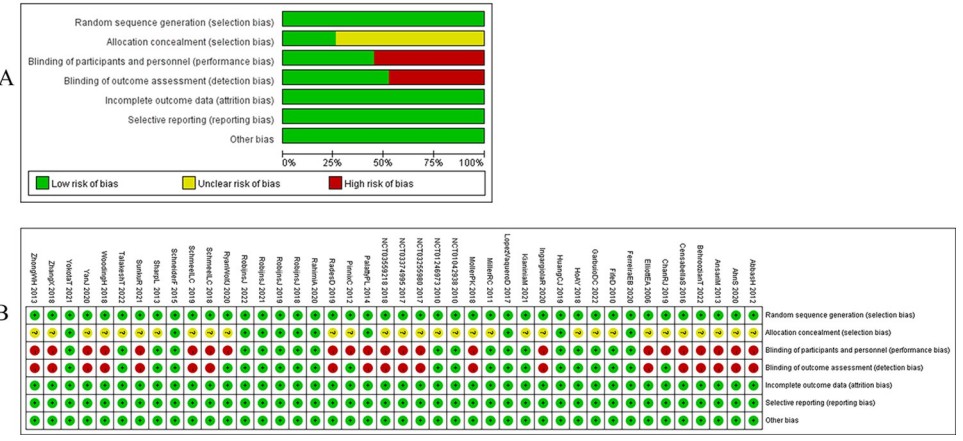

**Fig 2. Review authors' judgement for each risk of bias.** (A) Risk of bias graph, presented as percentage across all included studies; (B) Risk of bias summary, each risk of bias item for each included study.

exhibited a high risk of bias. The funnel plots for each outcome indicated that publication bias was under control. These funnel plots were presented in Fig 3.

## Network evaluations

Fig 4 illustrates the evidence network plots depicting the interventions tested in RCTs for their effectiveness in managing RD, focusing on primary outcomes. Similarly, Fig 5 presents the evidence network plots for secondary outcomes. In these plots, the size of the circles corresponds to the number of studies that examined each intervention, while the thickness of the lines represents the number of studies that compared the two connected interventions.

**Primary outcomes.** The incidence of grade≥2 RD was assessed using 13 pairwise comparisons, which involved 37 studies and 13 interventions compared to SOC. The interventions included in this analysis were StrataXRT®, Mepitelfilm, Trolamine, Hyaluronicacid, PBMT, Chamomile, Hydrofilm, Calendula, Curcumin, Glutamine, Xonrid®, Corticosteroid, and

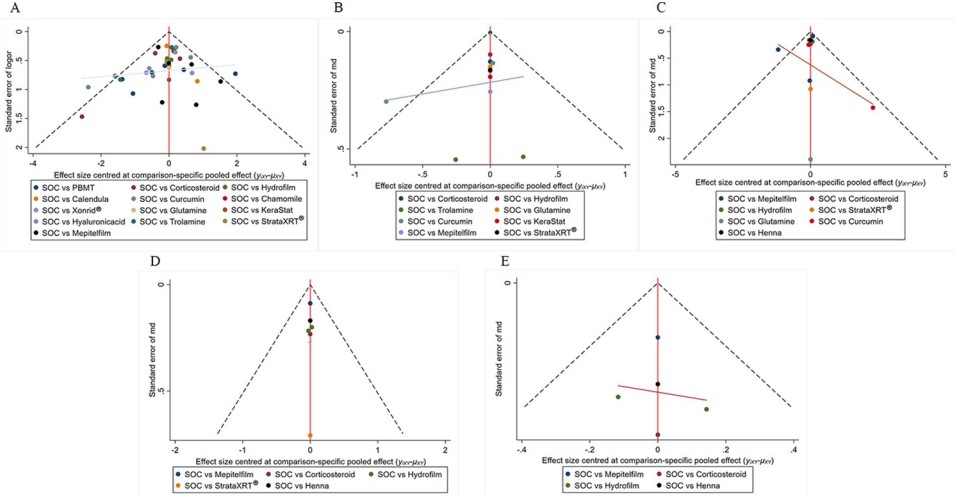

**Fig 3. The net-funnel graphs of primary and secondary outcomes.** (A) Incidence of grade≥2 RD, (B) RD sores, (C) Pain sensations, (D) Itching sensations, and (E) Burning sensations.

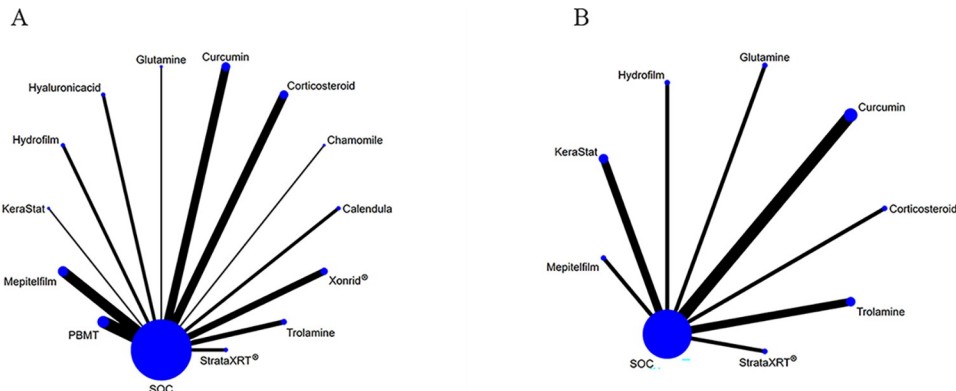

**Fig 4. Network plots of available interventions comparisons for primary outcomes.** (A) Incidence of grade≥2 RD, (B) RD sores. Line width is proportional to the number of trials including every pair of interventions (direct comparisons). Circle size is proportional to the total number of patients for each intervention in the network.

KeraStat (Fig 4A). The RD score was determined through 8 pairwise comparisons, which encompassed 11 studies and 8 interventions compared to SOC. The interventions included in this analysis were StrataXRT®, Mepitelfilm, Trolamine, Hydrofilm, Curcumin, Glutamine, Corticosteroid, and KeraStat.

**Secondary outcomes.** For pain sensations, there were 7 pairwise comparisons contributing to NMA, which involved 11 studies and 7 interventions compared to SOC. The interventions included in this analysis were StrataXRT®, Mepitelfilm, Hydrofilm, Curcumin, Glutamine, Corticosteroid, and Henna.

Regarding itching sensations, there were 5 pairwise comparisons that contributed to the NMA, involving 6 studies and 5 interventions compared to SOC. The interventions included in this analysis were StrataXRT®, Mepitelfilm, Hydrofilm, Corticosteroid, and Henna.

For burning sensations, there were 4 pairwise comparisons contributing to the NMA, involving 5 studies and 4 interventions compared to SOC. The interventions included in this analysis were Mepitelfilm, Hydrofilm, Corticosteroid, and Henna.

## Forest plots

Figs 6 and 7 display the findings of the direct pairwise meta-analyses comparing various interventions with SOC for both primary and secondary outcomes.

**Primary outcomes.** Fig 6 shows the full findings of forest plots of network meta-analysis of all trials for primary outcomes.

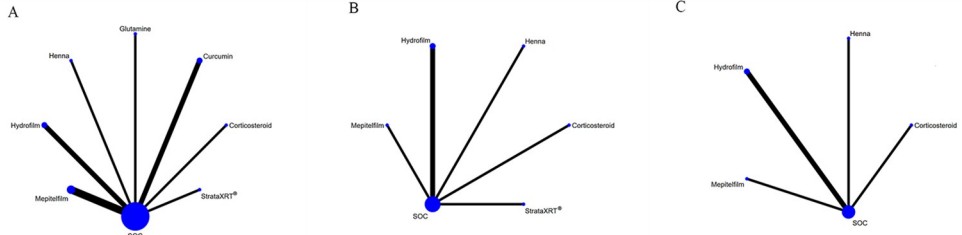

**Fig 5. Network plots of available interventions for secondary outcomes.** (A) Pain sensations, (B) Itching sensations, and (C) Burning sensations. Line width is proportional to the number of trials including every pair of interventions (direct comparisons). Circle size is proportional to the total number of patients for each intervention in the network.

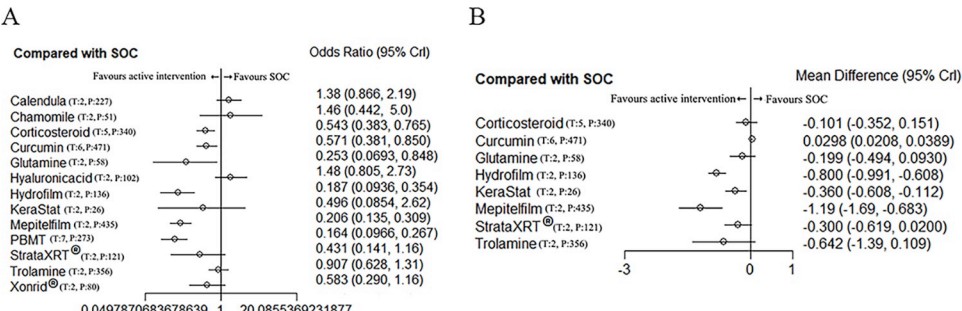

**Fig 6. Forest plots of network meta-analysis of all trials for primary outcomes.** (A) measured as odds ratio in incidence of grade≥2 RD, and (B) measured as mean difference for RD score. Interventions compared with SOC, which was the reference intervention. CrI = credible interval, T = number of trials, P = total number of patients. OR = odds ratio, SOC = standard of care.

ORs for the incidence of grade≥2 RD demonstrated a significant difference when compared to SOC. Specifically, Mepitelfilm (OR 0.21 [0.14–0.31]), PBMT (OR 0.16 [0.097–0.27]), Hydrofilm (OR 0.19 [0.09–0.35]), Curcumin (OR 0.57 [0.38–0.85]), Glutamine (OR 0.25 [0.07–0.85]), and Corticosteroid (OR 0.54 [0.38–0.77]) showed a more favorable outcome compared to SOC.

In terms of the RD score, Hydrofilm (MD-0.80 [-0.99– -0.61]), KeraStat (MD-0.36 [-0.61– -0.11]), and Mepitelfilm (MD-0.19 [-1.69– -0.68]) exhibited superior performance compared to SOC.

**Secondary outcomes.** Fig 7 shows the full findings of forest plots of network meta-analysis of all trials for secondary outcomes.

For pain sensations, Hydrofilm (MD-0.49 [-0.73– -0.25]), Henna (MD-0.93 [-1.25– -0.61]), and Mepitelfilm (MD-0.38 [-0.55– -0.22]) exhibited an advantage over SOC in terms of providing relief from pain.

Regarding itching sensations, Corticosteroid (MD-0.70 [-1.2– -0.25]), Hydrofilm (MD-0.70 [-0.99– -0.41]), Henna (MD-0.44 [-0.77– -0.11]), and Mepitelfilm (MD-0.20 [-0.37– -0.03]) demonstrated greater effectiveness compared to SOC in alleviating itching.

In relation to burning sensations, Corticosteroid (MD-0.60 [-1.1– -0.12]), Hydrofilm (MD-0.50 [-0.77– -0.23]), Henna (MD-0.46 [-0.78– -0.14]), and Mepitelfilm (MD-0.20 [-0.37– -0.03]) were found to be more helpful than SOC in reducing burning sensations.

## League tables

We conducted direct and indirect comparisons to assess the comparative effectiveness of different interventions using league tables (Tables 2–6). The direct comparisons align with the findings presented in the forest plots (Figs 6 and 7). Additionally, significant differences were observed among the interventions when considering mixed comparisons.

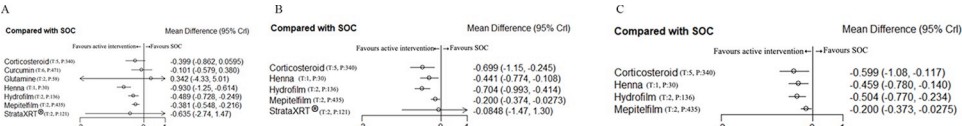

**Fig 7. Forest plots of network meta-analysis of all trials for secondary outcomes.** (A) measured as mean difference in pain sensations, (B) measured as mean difference for itching sensations, and (C) measured as mean difference for burning sensations. Interventions compared with SOC, which was the reference intervention. CrI = credible interval, T = number of trials, P = total number of patients, OR = odds ratio, SOC = standard of care.

**Table 2. League table with network meta-analysis (NMA) estimates for incidence of ≥Grade2 radiodermatitis (RD) (primary outcome).**

| Calendula | | | | | | | | | | | | | |
|---|---|---|---|---|---|---|---|---|---|---|---|---|---|
| **Chamomile** | | | | | | | | | | | | | |
| 0.95 (0.25, 3.39) | Chamomile | | | | | | | | | | | | |
| **2.53 (1.43, 4.54)** | 2.68 (0.77, 9.62) | Corticosteroid | | | | | | | | | | | |
| **2.41 (1.31, 4.47)** | 2.55 (0.73, 9.38) | 0.95 (0.56, 1.62) | Curcumin | | | | | | | | | | |
| **5.45 (1.48, 21.61)** | **5.79 (1.05, 34.32)** | 2.15 (0.61, 8.23) | 2.26 (0.63, 8.74) | Glutamine | | | | | | | | | |
| 0.93 (0.43, 2.01) | 0.99 (0.26, 3.91) | *0.37 (0.18, 0.74)* | 0.39 (0.18, 0.8) | 0.17 (0.04, 0.67) | Hyaluronicacid | | | | | | | | |
| **7.37 (3.35, 16.95)** | **7.82 (1.99, 32.07)** | **2.9 (1.41, 6.27)** | **3.06 (1.42, 6.8)** | 1.36 (0.32, 5.44) | **7.92 (3.27, 19.79)** | Hydrofilm | | | | | | | |
| 2.78 (0.49, 17.05) | 2.96 (0.38, 24.72) | 1.1 (0.2, 6.57) | 1.15 (0.21, 6.94) | 0.51 (0.06, 4.29) | 2.98 (0.51, 19.06) | 0.38 (0.06, 2.42) | KeraStat | | | | | | |
| **6.69 (3.6, 12.54)** | **7.09 (2.01, 25.96)** | **2.64 (1.55, 4.53)** | **2.78 (1.56, 4.94)** | 1.23 (0.32, 4.44) | **7.17 (3.46, 15.1)** | 0.91 (0.41, 1.96) | 2.41 (0.4, 13.38) | Mepitelfilm | | | | | |
| **8.41 (4.28, 16.93)** | **8.92 (2.46, 34.11)** | **3.32 (1.82, 6.2)** | **3.49 (1.84, 6.72)** | 1.55 (0.39, 5.83) | **9.04 (4.13, 20.1)** | 1.14 (0.49, 2.64) | 3.03 (0.49, 17.3) | 1.26 (0.66, 2.44) | PBMT | | | | |
| 1.38 (0.87, 2.19) | 1.46 (0.44, 5) | *0.54 (0.38, 0.77)* | *0.57 (0.38, 0.85)* | *0.25 (0.07, 0.85)* | 1.48 (0.81, 2.73) | *0.19 (0.09, 0.35)* | 0.5 (0.09, 2.62) | *0.21 (0.14, 0.31)* | *0.16 (0.1, 0.27)* | SOC | | | |
| **3.21 (1.07, 10.66)** | 3.41 (0.72, 17.74) | 1.26 (0.44, 4.03) | 1.33 (0.46, 4.35) | 0.59 (0.12, 3.07) | 3.44 (1.08, 12.21) | 0.43 (0.13, 1.57) | 1.16 (0.15, 8.53) | 0.48 (0.16, 1.56) | 0.38 (0.12, 1.28) | 2.32 (0.86, 7.11) | StrataXRT® | | |
| 1.52 (0.84, 2.75) | 1.6 (0.46, 5.82) | *0.6 (0.36, 0.99)* | 0.63 (0.36, 1.08) | 0.28 (0.07, 1) | 1.63 (0.8, 3.33) | *0.21 (0.09, 0.43)* | 0.55 (0.09, 2.99) | *0.23 (0.13, 0.39)* | *0.18 (0.09, 0.33)* | 1.1 (0.76, 1.59) | 0.47 (0.15, 1.37) | Trolamine | |
| **2.36 (1.03, 5.44)** | 2.5 (0.63, 10.24) | 0.93 (0.43, 2.03) | 0.98 (0.44, 2.19) | 0.43 (0.1, 1.77) | 2.53 (1.01, 6.4) | *0.32 (0.12, 0.83)* | 0.85 (0.13, 5.17) | *0.35 (0.16, 0.79)* | *0.28 (0.12, 0.66)* | 1.72 (0.86, 3.45) | 0.74 (0.2, 2.47) | 1.56 (0.71, 3.43) | Xonrid® |

Comparisons of incidence of ≥Grade2 RD of different managements were shown and should be read from left to right. The effectiveness estimate is located at the intersection of the column-defining treatment and the row-defining treatment. Incidence of ≥Grade2 RD estimates are presented in odds ratio (OR) with the 95% Credible Intervals (CrI), a OR below 1.0 favors the row-defining management (less presence of incidence of ≥Grade2 RD, means the top-left treatment is better). Significant results are in red with Bayesian $P<0.05$. Cells filling represent: Bold formatting: A OR of >1; Italic formatting and light gray: A OR of <1.

**Table 3. League table with network meta-analysis (NMA) estimates for radiodermatitis (RD) score (primary outcome).**

| Corticosteroid | Corticosteroid | | | | | | | | |
|---|---|---|---|---|---|---|---|---|---|
| Curcumin | -0.13 (-0.38, 0.12) | Curcumin | | | | | | | |
| Glutamine | 0.10 (-0.29, 0.49) | 0.23 (-0.06, 0.52) | Glutamine | | | | | | |
| Hydrofilm | **0.70 (0.38, 1.01)** | **0.83 (0.64, 1.02)** | **0.60 (0.25, 0.95)** | Hydrofilm | | | | | |
| KeraStat | 0.26 (-0.09, 0.61) | **0.39 (0.14, 0.64)** | 0.16 (-0.22, 0.55) | *-0.44 (-0.75, -0.13)* | KeraStat | | | | |
| Mepitelfilm | **1.09(0.52, 1.65)** | **1.22 (0.71, 1.72)** | **0.99 (0.40, 1.57)** | **0.39 (-0.15, 0.93)** | 0.83 (0.26, 1.39) | Mepitelfilm | | | |
| SOC | -0.10 (-0.35, 0.15) | **0.03 (0.02, 0.04)** | -0.20 (-0.49, 0.09) | *-0.80 (-0.99, -0.61)* | *-0.36 (-0.61, -0.11)* | *-1.19 (-1.69, -0.68)* | SOC | | |
| StrataXRT® | 0.20 (-0.21, 0.61) | **0.33 (0.01, 0.65)** | 0.10 (-0.34, 0.54) | *-0.50 (-0.87, -0.13)* | **-0.06 (-0.46, 0.34)** | *-0.89 (-1.49, -0.29)* | 0.30 (-0.02, 0.62) | StrataXRT® | |
| Trolamine | 0.54 (-0.25, 1.32) | 0.67 (-0.08, 1.42) | 0.44 (-0.36, 1.25) | -0.16 (-0.93, 0.61) | 0.28 (-0.51, 1.07) | -0.55 (-1.45 0.35) | 0.64 (-0.11, 1.39) | 0.34 (-0.48, 1.15) | **Trolamine** |

Comparisons of RD score of different managements were shown and should be read from left to right. The effectiveness estimate is located at the intersection of the column-defining treatment and the row-defining treatment. RD score estimates are presented in mean difference (MD) with the 95% Credible Intervals (CrI), a MD below zero favors the row-defining management (less assessment of RD score, means the top-left treatment is better). Significant results are in red with Bayesian $P<0.05$. Cells filling represent: Bold formatting: A MD of $>0$; Italic formatting and light gray: A MD of $<0$.

**Primary outcomes.** In terms of the incidence of grade≥2 RD (Table 2), Corticosteroid, Curcumin, and Glutamine demonstrated greater effectiveness compared to Hyaluronicacid, with ORs of (OR 0.37 [0.18, 0.74]), (OR 0.39 [0.18, 0.8]), and (OR 0.17 [0.04, 0.67]), respectively. Additionally, Corticosteroid, Hydrofilm, Mepitelfilm, and PBMT were found to be superior to Trolamine, with ORs of (OR 0.6 [0.36, 0.99]), (OR 0.21 [0.09, 0.43]), (OR 0.23 [0.13, 0.39]), and (OR 0.18 [0.09, 0.33]), respectively. Furthermore, Hydrofilm, Mepitelfilm,

**Table 4. League table with network meta-analysis (NMA) estimates for pain sensations (secondary outcome).**

| Corticosteroid | Corticosteroid | | | | | | | |
|---|---|---|---|---|---|---|---|---|
| Curcumin | -0.30 (-0.96, 0.36) | Curcumin | | | | | | |
| Glutamine | -0.75 (-5.44, 3.95) | -0.45 (-5.13, 4.26) | Glutamine | | | | | |
| Henna | 0.53 (-0.03, 1.09) | **0.83 (0.26, 1.41)** | 1.27 (-3.41, 5.96) | Henna | | | | |
| Hydrofilm | 0.09 (-0.43, 0.61) | 0.39 (-0.15, 0.922) | 0.83 (-3.85, 5.52) | *-0.44 (-0.84, -0.04)* | Hydrofilm | | | |
| Mepitelfilm | -0.02 (-0.51, 0.47) | 0.28 (-0.22, 0.79) | 0.73 (-3.95, 5.40) | *-0.55 (-0.91, -0.19)* | -0.11 (-0.40, 0.18) | Mepitelfilm | | |
| SOC | -0.40 (-0.86, 0.06) | -0.10 (-0.58, 0.38) | 0.34 (-4.33, 5.01) | *-0.93 (-1.25, -0.61)* | *-0.49 (-0.73, -0.25)* | *-0.38 (-0.55, -0.22)* | SOC | |
| StrataXRT® | 0.24 (-1.91, 2.39) | 0.54 (-1.63, 2.69) | 0.98 (-4.17, 6.12) | -0.29 (-2.42, 1.84) | 0.15 (-1.97, 2.26) | 0.25 (-1.86, 2.37) | 0.63 (-1.47, 2.74) | StrataXRT® |

Comparisons of pain sensations of different managements were shown and should be read from left to right. The effectiveness estimate is located at the intersection of the column-defining treatment and the row-defining treatment. Pain sensations estimates are presented in mean difference (MD) with the 95% Credible Intervals (CrI), a MD below zero favors the row-defining management (more effective pain relief, means the top-left treatment is better). Significant results are in red with Bayesian $P<0.05$. Cells filling represent: Bold formatting: A MD of $>0$; Italic formatting and light gray: A MD of $<0$.

**Table 5. League table with network meta-analysis (NMA) estimates for itching sensations (secondary outcome).**

| Corticosteroid | Corticosteroid | | | | |
|---|---|---|---|---|---|
| **Henna** | -0.26 (-0.82, 0.31) | **Henna** | | | |
| **Hydrofilm** | 0.00 (-0.53, 0.54) | 0.26 (-0.18, 0.70) | **Hydrofilm** | | |
| **Mepitelfilm** | *-0.50(-0.98, -0.01)* | -0.24 (-0.62, 0.13) | *-0.50 (-0.84, -0.17)* | **Mepitelfilm** | |
| **SOC** | *-0.70 (-1.15, -0.24)* | *-0.44 (-0.77, -0.11)* | *-0.70 (-0.99, -0.41)* | *-0.20 (-0.37, -0.03)* | **SOC** |
| **StrataXRT®** | -0.61 (-2.07, 0.85) | -0.362 (-1.77, 1.07) | -0.62 (-2.03, 0.80) | -0.12 (-1.51, 1.28) | 0.08 (-1.30, 1.47) | **StrataXRT®** |

Comparisons of itching sensations of different managements were shown and should be read from left to right. The effectiveness estimate is located at the intersection of the column-defining treatment and the row-defining treatment. Itching sensations estimates are presented in mean difference (MD) with the 95% Credible Intervals (CrI), a MD below zero favors the row-defining management (more helpful itching relief, means the top-left treatment is better). Significant results are in red with Bayesian $P<0.05$. Cells filling represent: Bold formatting: A MD of $>0$; Italic formatting and light gray: A MD of $<0$.

**Table 6. League table with network meta-analysis (NMA) estimates for burning sensations (secondary outcome).**

| Corticosteroid | Corticosteroid | | | |
|---|---|---|---|---|
| **Henna** | -0.14 (-0.72, 0.44) | **Henna** | | |
| **Hydrofilm** | -0.09 (-0.65, 0.46) | 0.04 (-0.37, 0.46) | **Hydrofilm** | |
| **Mepitelfilm** | -0.40(-0.91, 0.11) | -0.26 (-0.62, 0.10) | -0.30 (-0.62, 0.01) | **Mepitelfilm** |
| **SOC** | *-0.60 (-1.08, -0.12)* | *-0.46 (-0.78, -0.14)* | *-0.50 (-0.77, -0.23)* | *-0.20 (-0.37, -0.03)* | **SOC** |

Comparisons of burning sensations of different managements were shown and should be read from left to right. The effectiveness estimate is located at the intersection of the column-defining treatment and the row-defining treatment. Burning sensations estimates are presented in mean difference (MD) with the 95% Credible Intervals (CrI), a MD below zero favors the row-defining management (more helpful burning relief, means the top-left treatment is better). Significant results are in red with Bayesian $P<0.05$. Cells filling represent: Bold formatting: A MD of $>0$; Italic formatting and light gray: A MD of $<0$.

Mepitelfilm = Mepitel film, Mepilex Lite

Trolamine = Trolamine, Trolamine cream

Hyaluronicacid = Hyaluronic acid, HÀ formulation

PBMT = MLS® laser therapy (LT), Light emitting diode (LED) photomodulation, Photobiomodulation therapy, Red light phototherapy (RLPT)

Chamomile = Chamomile gel

Calendula = Calendula cream

Curcumin = Nanocurcumin, Curcumin C3 complex, Curcumin cream, Vicco® turmeric cream (VTC)

Henna = Alpha ointment

Glutamine = Oral glutamine, L- Glutamine

Corticosteroid = Difluprednate (MYSER ointment 0.05%), Mometasone, 0.1% mometasone furoate, Betamethasone0.1%

KeraStat = KeraStat(R) Cream, Keratin

SOC = Standard of care, including: a) Standard skin care, standard of care, standard care, placebo, control, usual care, sham treatment, institutional preference or no intervention; b) Institutional nursing staffs used the currently recommended standard skin care treatment and standard products for prevention and treatment of these skin lesions as a part of hospital protocols or institutional standard for the trials such as hydrocortisone, urea cream, HPRPlusTM, Biafine® cream, white Vaseline (PROPETO), Eucerin Original® (E) cream and Sorbolene® cream, as well as moisturizer/moisturizing base cream, aqueous creams, and other conventional products or normal care.

and PBMT exhibited better performance than Xonrid®, with ORs of (OR 0.32 [0.12, 0.83]), (OR 0.35 [0.16, 0.79]), and (OR 0.28 [0.12, 0.66]), respectively. For RD score (Table 3), Hydrofilm was identified as the superior intervention for reducing the RD score. MDs for Hydrofilm compared to other interventions (StrataXRT® and KeraStat) ranged from −0.50 to −0.44. Additionally, Mepitelfilm showed notable effectiveness with an MD of -0.89 when compared to StrataXRT®.

**Secondary outcomes.** For pain sensations (Table 4), Henna was ranked as the most effective intervention for pain relief. MDs for Henna compared to Mepitelfilm and Hydrofilm ranged from −0.55 to −0.44.

In terms of itching sensations (Table 5), Corticosteroid and Hydrofilm exhibited similar efficacy to Mepitelfilm, with MDs of -0.50 (-0.98, -0.01) and -0.50 (-0.84, -0.17), respectively.

However, for burning sensations (Table 6), Corticosteroid, Hydrofilm, Mepitelfilm, and Henna did not yield significantly different outcomes when compared to each other.

### Ranking of interventions by efficacy

The ranking probability of different interventions in managing RD is presented in Table 7 and Figs 8–12. Based on the SUCRA plot and values, PBMT (SUCRA, 0.92) achieved the highest ranking in terms of the lowest incidence of grade≥2 RD, while Hyaluronicacid (SUCRA, 0.10) obtained the lowest ranking (Fig 8). In terms of decreasing RD scores, Mepitelfilm (SUCRA, 0.98) ranked the highest, while Curcumin (SUCRA, 0.03) ranked the lowest (Fig 9). Henna (SUCRA, 0.89) was ranked the highest for pain sensations relief, whereas SOC (SUCRA, 0.17) received the lowest ranking (Fig 10). Regarding the relief of itching sensations, Hydrofilm (SUCRA, 0.84) demonstrated the greatest efficacy in reducing itching sensations, while SOC (SUCRA, 0.09) had the highest incidence of itching sensations (Fig 11). For alleviating burning sensations, Corticosteroid (SUCRA, 0.81) exhibited the most effective results, while SOC (SUCRA, 0.01) was the least effective intervention (Fig 12).

## Discussion

In this study, we conducted a Bayesian NMA to evaluate the efficacy of different interventions used for the prevention and treatment of RD. Consistent use of assessment tools is important to document the severity of RD and respond with appropriate therapeutic interventions. Clinicians often rely on clinician-reported outcomes (CROs) to assess the incidence of grade≥2 RD, specifically the percentage of moist desquamation, and RD score. These measures are commonly utilized as primary outcome measures in RD-related systematic reviews and clinical trials. They offer quantitative and standardized assessments of RD severity, allowing for meaningful comparisons across different studies. By considering both the analysis of the incidence of grade≥2 RD and RD score, the impact of a particular intervention in reducing the severity of RD among patients is evaluated comprehensively.

Based on our cumulative probability ranking, PBMT demonstrated significantly better outcomes compared to Trolamine and Xonrid® in reducing the incidence of grade≥2 RD. PBMT is a non-invasive and a-thermal treatment approach that utilizes light to activate epithelial healing through various metabolic processes involving photochemical reactions [52]. PBMT offers advantages such as enhanced light penetration and propagation within tissues, leading to greater anti-inflammatory, bio-stimulating, and analgesic effects. Several open-label studies have provided evidence of PBMT's effectiveness in preventing the development of grade 2 or higher acute RD [48, 50–52]. Despite this evidence, PBMT is not currently recommended for clinical use by the Society and College of Radiographers (SCoR) due to uncertainty regarding the replicability of study results. In 2010, Fife, D. presented an opposing viewpoint

**Table 7. The SUCRA probabilities of different interventions for acute RD on clinical effectiveness.**

| Intervention Effect | SOC | Calendula | Chamomile | Corticosteroid | Curcumin | Glutamine | Henna | Hyaluronicacid | Hydrofilm | KeraStat | Mepitelfilm | PBMT | StrataXRT® | Trolamine | Xonrid® |
|---|---|---|---|---|---|---|---|---|---|---|---|---|---|---|---|
| Incidence of ≥Grade2 RD | 0.24 | 0.11 | 0.15 | 0.54 | 0.51 | 0.77 | NA | 0.10 | 0.88 | 0.54 | 0.85 | **0.92** | 0.61 | 0.30 | 0.50 |
| RD scores | 0.17 | NA | NA | 0.28 | 0.03 | 0.40 | NA | NA | 0.84 | 0.57 | **0.98** | NA | 0.51 | 0.71 | NA |
| Pain sensations | 0.17 | NA | NA | 0.53 | 0.28 | 0.37 | **0.89** | NA | 0.63 | NA | 0.51 | NA | 0.60 | NA | NA |
| Itching sensations | 0.09 | NA | NA | 0.82 | NA | NA | 0.58 | NA | **0.84** | NA | 0.34 | NA | 0.34 | NA | NA |
| Burning sensations | 0.01 | NA | NA | **0.81** | NA | NA | 0.66 | NA | 0.73 | NA | 0.29 | NA | NA | NA | NA |

RD = Radiodermatitis

NA = Not available

SUCRA = Value of surface under the cumulative ranking curve (The higher the SUCRA value, the higher possible ranking was that of the intervention.)

Mepitelfilm = Mepitel film, Mepilex Lite

Trolamine = Trolamine, Trolamine cream

Hyaluronicacid = Hyaluronic acid, HA formulation

PBMT = MLS® laser therapy (LT), Light emitting diode (LED) photomodulation, Photobiomodulation therapy, Red light phototherapy (RLPT)

Chamomile = Chamomile gel

Calendula = Calendula cream

Curcumin = Nanocurcumin, Curcumin C3 complex, Curcumin cream, Vicco® turmeric cream (VTC)

Henna = Alpha ointment

Glutamine = Oral glutamine, L- Glutamine

Corticosteroid = Difluprednate (MYSER ointment 0.05%), Mometasone, 0.1% mometasone furoate, Betamethasone0.1%

KeraStat = KeraStat(R) Cream, Keratin

SOC = Standard of care, including: a) Standard skin care, standard of care, standard care, placebo, control, usual care, sham treatment, institutional preference or no intervention; b) Institutional nursing staffs used the currently recommended standard skin care treatment and standard products for prevention and treatment of these skin lesions as a part of hospital protocols or institutional standard for the trials such as hydrocortisone, urea cream, HPRPlusTM, Biafine® cream, white Vaseline (PROPETO), Eucerin Original® (E) cream and Sorbolene® cream, as well as moisturizer/ moisturizing base cream, aqueous creams, and other conventional products or normal care.

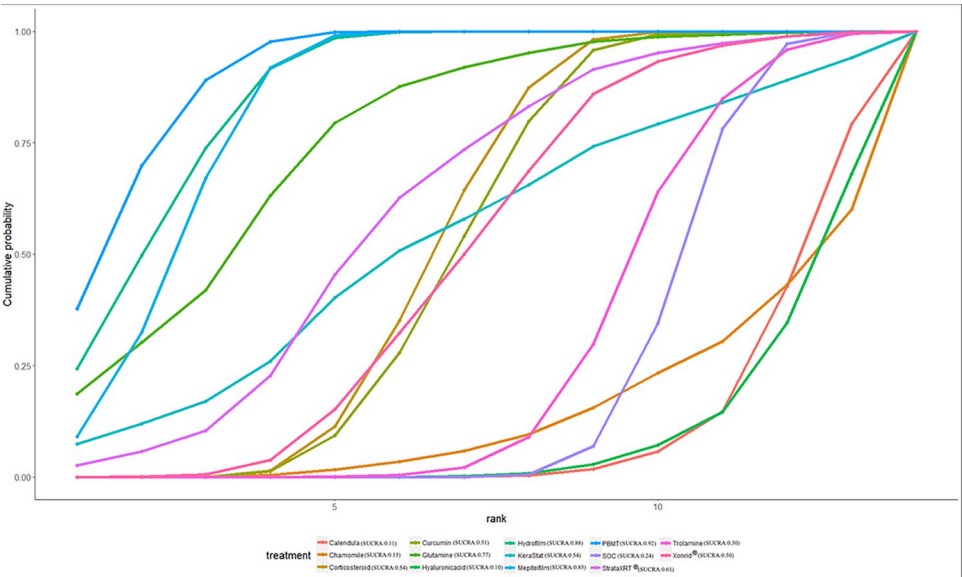

**Fig 8. The surface under curve cumulative ranking area (SUCRA) of incidence of grade≥2 RD for effective outcomes.** The SUCRA value would be 0 when an intervention is certain to be the worst and 1 when it is certain to be the best.

suggesting that PBMT may not effectively reduce the severity of skin reactions, indicating that the effectiveness of PBMT may be influenced by various patient and treatment-related factors [49].

Many scoring systems used to assess RD share several similarities. Two similarities are as follows: (1) These scoring systems assess the severity of radiation dermatitis based on specific criteria, including erythema, edema, desquamation, and pain; and (2) these scoring systems utilize a grading system to categorize the severity of radiation dermatitis, typically ranging

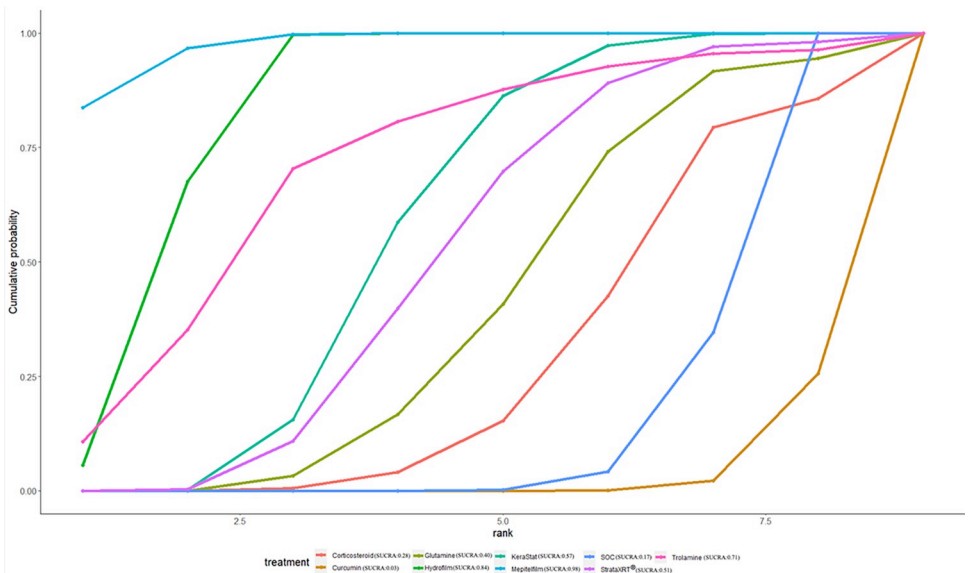

**Fig 9. The surface under curve cumulative ranking area (SUCRA) of RD sores for effective outcomes.** The SUCRA value would be 0 when an intervention is certain to be the worst and 1 when it is certain to be the best.

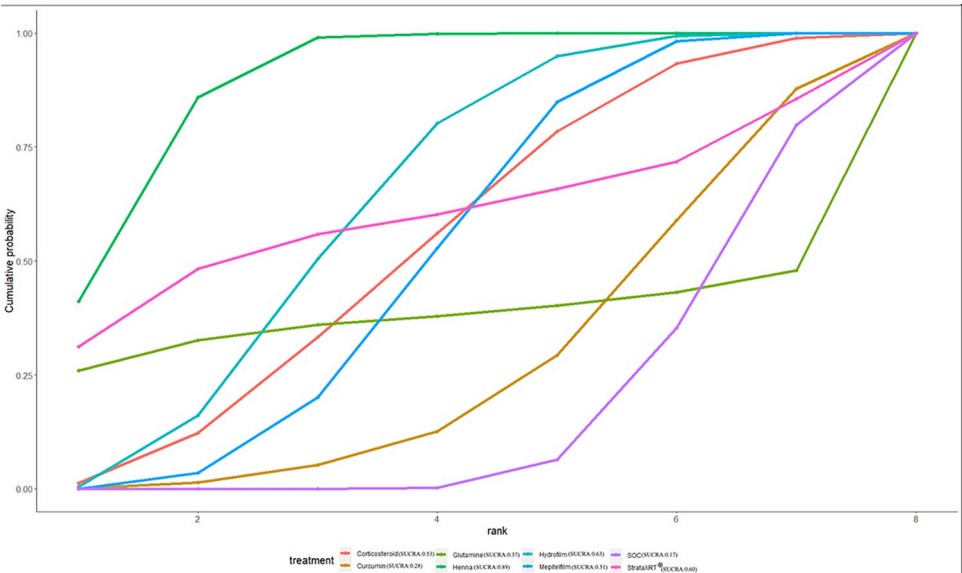

**Fig 10. The surface under curve cumulative ranking area (SUCRA) of pain sensations for effective outcomes.** The SUCRA value would be 0 when an intervention is certain to be the worst and 1 when it is certain to be the best.

from grade 1 (mild) to grade 4 (severe). In terms of RD score, Mepitelfilm was more effective than other interventions in our findings, which was in agreement with several own-control studies [38, 40–42]. Mepitelfilm is a sterile, transparent, breathable, and adhesive soft silicone film. The utilization of barrier protectants is grounded in a fundamental concept and a potential mechanism that assumes safeguarding the keratinized surface and shielding the radiotherapy-affected basal stem-cell layer from surface abrasion and friction can significantly prevent skin damage. Additionally, a systematic review and meta-analysis conducted by Robijns et al.

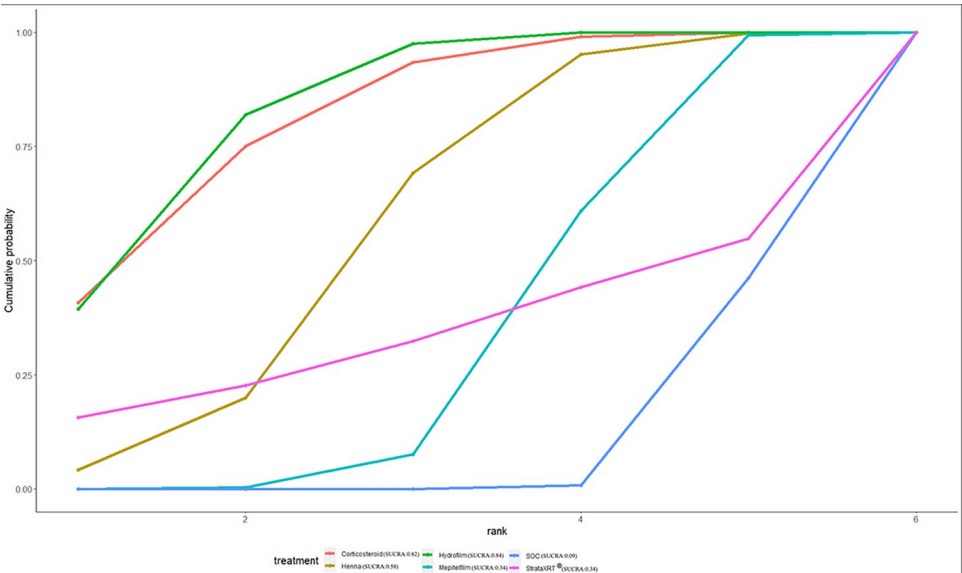

**Fig 11. The surface under curve cumulative ranking area (SUCRA) of itching sensations for effective outcomes.** The SUCRA value would be 0 when an intervention is certain to be the worst and 1 when it is certain to be the best.

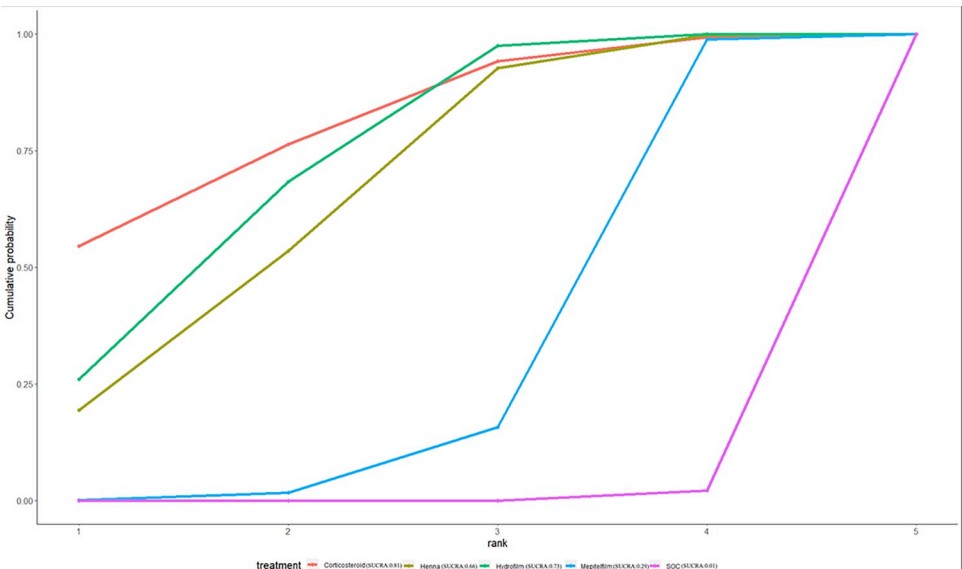

**Fig 12. The surface under curve cumulative ranking area (SUCRA) of burning sensations for effective outcomes.**
The SUCRA value would be 0 when an intervention is certain to be the worst and 1 when it is certain to be the best.

[7] provided evidence showing that Mepitelfilm had a positive impact on the severity of radiation dermatitis. Their subgroup analysis revealed that Mepitelfilm significantly reduced grade 2 radiation dermatitis in head and neck cancer patients, but did not affect the incidence of moist desquamation in these patients. Mepitelfilm was found to be more effective for radiation dermatitis of grade 2 or higher in breast cancer patients. Meanwhile, the use of barrier films and dressings lacks widespread consensus or strong recommendations among guidelines due to the challenges associated with the difficulty of designing unbiased studies on this topic. Until now, most studies on RD have relied on CROs as the primary measure for guiding clinical practice and symptom management, considering them the "gold standard." However, there has been a growing focus on incorporating the patient perspective in recent RD studies [80]. Based on the SUCRA outcome, Henna demonstrated the highest ranking in terms of improving pain relief, showing significant differences. Henna is derived from natural compounds found in Lawsonia inermis, and some researchers have presented evidence supporting its antimicrobial and antioxidant properties in the context of wound healing [75]. So far, no recommendation was made for any natural agent by practice guidelines. Our research provided valuable insights into the potential of novel herbal interventions that specifically target pain relief. In terms of PROs, Hydrofilm emerged as the intervention with the lowest incidence of itching sensations reported by patients. Hydrofilm is a transparent, waterproof polyurethane film that is coated with a hypoallergenic polyacrylate adhesive. Our findings corroborate the results of previous meta-analyses conducted by Robijns et al. [7]. Analyzing the pooled data, we found that Hydrofilm significantly reduced the average itching score and had a positive impact on limiting day-to-day activities. However, our analysis did not find a significant effect of Hydrofilm on the mean pain and burning sensation scores.

Regarding the evaluation of burning sensations, Corticosteroid proved to be the most beneficial treatment, although its effectiveness did not show statistical significance when compared to other interventions. RD is linked to an inflammatory process believed to be mediated by cytokines. Corticosteroids function as anti-inflammatory substances by controlling the attachment of white blood cells to endothelial cells, causing blood vessel constriction, reducing the

permeability of capillaries, and inhibiting the multiplication and movement of white blood cells. In laboratory studies, they have been demonstrated to decrease the production of interleukin 6 [6]. Currently, most guidelines recommend the use of hydrocortisone, a moderately potent corticosteroid, in the usual institutional care to reduce the occurrence of grade≥2 RD. This recommendation is based on the lack of clear evidence supporting the superiority of other potent corticosteroids over hydrocortisone in the literature. However, it is important to note a caveat regarding its use. Prolonged use of hydrocortisone carries a significant risk of skin atrophy, telangiectasia, infection, and impaired wound healing. These factors limit its acceptance as a prescribed treatment during radiation therapy. In the current study, several newly developed high to medium potency corticosteroids, such as difluprednate, mometasone furoate, and betamethasone, were tested. In an RCT conducted by Miller RC et al. [71], utilizing three tools to measure PROs, the group receiving corticosteroids reported less itching and a lower persistence or recurrence of burning symptoms. These findings align with our own analysis. Therefore, according to the ISNCC (International Society of Nurses in Cancer Care, 2021), betamethasone valerate cream and mometasone furoate cream are recommended for managing RD during RT. However, it is emphasized that discontinuation of these creams is necessary if the skin becomes disrupted [17]. On the other hand, Ho, A. Y. et al. [69] reported no significant difference in patient-reported skin outcomes between the treatment arms (60 or 64 patients each treatment arm). Therefore, when comparing the effectiveness of different interventions, it is important to consider not only the results shown in the network plot but also the number of trials and participants involved, as well as the evidence from direct comparisons. Additionally, it is worth noting that patients may have reported more severe pain and other side effects compared to clinicians, possibly due to differences in cultural background, individual pain tolerance, and anticipation of symptom severity. These factors may partly explain the discrepancy in the reported scores.

This study has several notable strengths. Firstly, it successfully created the first network that compared 14 contemporary interventions for RD. Additionally, flexible Bayesian hierarchical modeling was employed, allowing for the integration of both direct and indirect evidence, effectively harnessing the collective strength of multiple trials and providing conservative estimations and conclusions through shrinkage estimation. The robustness of the results was further emphasized through sensitivity analyses that explored different model prior specifications and inclusion criteria.

Nonetheless, this study is also subject to several limitations. Firstly, as with any meta-analysis, there is heterogeneity in the design and reporting of the trials included in the analysis. The assessment scales for grading RD and PROs were not consistently uniform across the studies. Secondly, only RCTs were included in this network meta-analysis. While RCTs are generally well-designed and provide high-quality clinical data, the number of studies available for inclusion may have been limited. Although the intervention network was successfully established, most direct comparisons relied on data from a single RCT connection, resulting in relatively sparse direct evidence and limited assessment of incoherence. Thirdly, the outcomes would be improved if they were coupled with data from other common terms for RD such as skin reactions or skin burns in literature search. But it should be highlighted that Mesh and entry terms were used as search the most popular strategies. Fourthly, less than 30% of the trials included in the analysis were large studies. This could introduce bias due to the potential influence of small study effects. Fifthly, this meta-analysis was conducted based on aggregated study-level data. Individual patient characteristics, which can have a significant impact on treatment outcomes, were not accounted for and would require further investigation in future studies. Sixthly, while the primary focus of this study was the incidence of ≥Grade2 RD and RD scores as efficacy endpoints, adverse event (AE) data were not compared between interventions due

to incomplete or unavailable data in the included studies. Due to the majority of AE reports being mild or showing no response, occasional serious AEs are typically attributed to the overall treatment for cancer, and it becomes difficult to determine if they are specifically related to the intervention being studied. Seventhly, the SUCRA curve was utilized to estimate the ranking probability of comparative effectiveness among the various therapies. However, it is important to note that the SUCRA curve has its limitations, and the results should be interpreted with caution.

## Conclusion

Overall, this NMA provides a summary of the efficacy and impact on PROs for 14 different interventions used in the management of RD. Based on the findings, PBMT appears to have demonstrated better efficacy in reducing the incidence of grade≥2 RD, while Mepitelfilm showed better efficacy in reducing RD scores. When considering PROs, Henna treatment resulted in fewer complaints related to pain, while Hydrofilm showed fewer complaints related to itching sensations. When comparing different treatments, it is important to take into account the limitations of the available data, the characteristics of the patient group, and any uncertainties arising from the dosage selection or the specific context of therapy. However, since there is a lack of multi-arm RCTs that encompass all commonly used regimens, it is expected that more RCTs evaluating various interventions for RD will be conducted in the future. Despite these considerations, this NMA serves as a valuable and informative resource to guide treatment decisions in RD.

## Supporting information

**S1 Checklist. PRISMA-P (Preferred Reporting Items for Systematic review and Metanalysis Protocols) 2020 checklist: Recommended items to address in a systematic review protocol.**
(DOCX)

**S1 Appendix. Search strategy.**
(DOCX)

**S1 File. Included interventions in the Network meta-analysis by therapeutic class.**
(DOCX)

## Acknowledgments

The interpretation and reporting of these data are the authors' sole responsibility.

## Author Contributions

**Conceptualization:** Ying Guan, Shuai Liu.

**Data curation:** Ying Guan, Shuai Liu, Anchuan Li, Wanqin Cheng.

**Formal analysis:** Ying Guan, Shuai Liu.

**Funding acquisition:** Ying Guan.

**Investigation:** Ying Guan, Shuai Liu, Anchuan Li, Wanqin Cheng.

**Methodology:** Ying Guan, Shuai Liu.

**Project administration:** Ying Guan.

**Resources:** Ying Guan.

**Software:** Ying Guan.

**Supervision:** Shuai Liu, Anchuan Li, Wanqin Cheng.

**Validation:** Ying Guan, Shuai Liu.

**Visualization:** Ying Guan.

**Writing – original draft:** Ying Guan.

**Writing – review & editing:** Shuai Liu, Anchuan Li, Wanqin Cheng.

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
