## [Decision Letter · Decision Letter 0]

2 Nov 2023

PONE-D-23-26099Comparison of the efficacy among different interventions for radiodermatitis: a Bayesian network meta-analysis of randomized controlled trialsPLOS ONE

Dear Dr. Guan,

Thank you for submitting your manuscript to PLOS ONE. After careful consideration, we feel that it has merit but does not fully meet PLOS ONE’s publication criteria as it currently stands. Therefore, we invite you to submit a revised version of the manuscript that addresses the points raised during the review process.

Please submit your revised manuscript by Dec 17 2023 11:59PM.  If you will need more time than this to complete your revisions, please reply to this message or contact the journal office at plosone@plos.org. Please include the following items when submitting your revised manuscript:A rebuttal letter that responds to each point raised by the academic editor and reviewer(s). You should upload this letter as a separate file labeled 'Response to Reviewers'.A marked-up copy of your manuscript that highlights changes made to the original version. You should upload this as a separate file labeled 'Revised Manuscript with Track Changes'.An unmarked version of your revised paper without tracked changes. You should upload this as a separate file labeled 'Manuscript'.If applicable, we recommend that you deposit your laboratory protocols in protocols.io to enhance the reproducibility of your results. Protocols.io assigns your protocol its own identifier (DOI) so that it can be cited independently in the future. For instructions see: https://journals.plos.org/plosone/s/submission-guidelines#loc-laboratory-protocols. Additionally, PLOS ONE offers an option for publishing peer-reviewed Lab Protocol articles, which describe protocols hosted on protocols.io. Read more information on sharing protocols at https://plos.org/protocols?utm_medium=editorial-email&utm_source=authorletters&utm_campaign=protocols.

We look forward to receiving your revised manuscript.

Kind regards,

Jingxian Ding, Ph.D.

Academic Editor

PLOS ONE

Journal Requirements:

Additional Editor Comments:

Reviewer 1：

This study is a Bayesian network meta-analysis of various interventions for the management of radiodermatitis. Despite being the most common side-effect of radiotherapy, the optimal management strategy for radiodermatitis remains elusive. The authors have used a network meta-analysis (NMA) approach to rank different treatment strategies in terms of both clinician and patient-reported outcomes. The use of NMA is appropriate in this scenario and the study was conducted in accordance with the established guidelines for conducting systematic reviews and meta-analyses. The results of this study will offer clinicians much-needed information to make evidence-based decisions in managing radiodermatitis.

However, the authors need to address the following issues;

1. The authors make conflicting statements about the presence of publication bias in the quality assessment and results sections.

2. Under study selection and data collection, the authors report that Any disagreements were resolved through “team discussions”. More clarification should be provided on this.

3. In their literature search, the authors did not use other common terms for RD such as skin reactions or skin burns. This could result in failure to retrieve some eligible studies.

4. The authors can improve the clarity and readability of the manuscript by addressing the following issues;

• The pathogenesis of RD should be written in a less speculative way with reference to specific dermal components that are damaged by IR as well as involved signaling molecules. Also, authors should cite their sources here.

• The discussion is too long and difficult for the reader to follow. The authors should make it more concise and limit the discussion to results of their study.

• The description of NMA in the background should be condensed to one paragraph. Also, the authors should provide a preview of the risk factors for RD can provide a better context to the problem.

• There are several instances where the same information is repeated in the background, methodology, and discussion. The authors should rectify this.

5. The authors should ensure appropriate sources are cited. For instance, the opening statements in the background are not supported by any citations.

6. In tables 2B-2E, the authors report too many decimal places in the mean differences and 95% credible intervals.

7. In Figure 8, the labels are too small and some colors chosen look very similar.

8. There are several spelling and grammatical errors. The authors are advised to proofread their manuscript before re-submission.

Reviewer 2：

This is a meticuously worked out and written manuscript. It is very well written and quite detailed systematic review on prosepctive trials investigating treatments of radiation dermatitis. Conclusions are clear.

There is only thing, I would ask to adapt, change: there are two references in Brazilian, references number 13 and 14. Authors give non-English publications as an exclusion criterion. They should therefore be removed or substituted by other references. There are good English publications on the use of Calendula.

Reviewers' comments:

Reviewer's Responses to Questions

**Comments to the Author**

1. Is the manuscript technically sound, and do the data support the conclusions?

Reviewer #1: Yes

Reviewer #2: Yes

2. Has the statistical analysis been performed appropriately and rigorously? 

Reviewer #1: Yes

Reviewer #2: I Don't Know

3. Have the authors made all data underlying the findings in their manuscript fully available?

Reviewer #1: Yes

Reviewer #2: Yes

4. Is the manuscript presented in an intelligible fashion and written in standard English?

Reviewer #1: No

Reviewer #2: Yes

5. Review Comments to the Author

Reviewer #1: This study is a Bayesian network meta-analysis of various interventions for the management of radiodermatitis. Despite being the most common side-effect of radiotherapy, the optimal management strategy for radiodermatitis remains elusive. The authors have used a network meta-analysis (NMA) approach to rank different treatment strategies in terms of both clinician and patient-reported outcomes. The use of NMA is appropriate in this scenario and the study was conducted in accordance with the established guidelines for conducting systematic reviews and meta-analyses. The results of this study will offer clinicians much-needed information to make evidence-based decisions in managing radiodermatitis.

However, the authors need to address the following issues;

1. The authors make conflicting statements about the presence of publication bias in the quality assessment and results sections.

2. Under study selection and data collection, the authors report that Any disagreements were resolved through “team discussions”. More clarification should be provided on this.

3. In their literature search, the authors did not use other common terms for RD such as skin reactions or skin burns. This could result in failure to retrieve some eligible studies.

4. The authors can improve the clarity and readability of the manuscript by addressing the following issues;

• The pathogenesis of RD should be written in a less speculative way with reference to specific dermal components that are damaged by IR as well as involved signaling molecules. Also, authors should cite their sources here.

• The discussion is too long and difficult for the reader to follow. The authors should make it more concise and limit the discussion to results of their study.

• The description of NMA in the background should be condensed to one paragraph. Also, the authors should provide a preview of the risk factors for RD can provide a better context to the problem.

• There are several instances where the same information is repeated in the background, methodology, and discussion. The authors should rectify this.

5. The authors should ensure appropriate sources are cited. For instance, the opening statements in the background are not supported by any citations.

6. In tables 2B-2E, the authors report too many decimal places in the mean differences and 95% credible intervals.

7. In Figure 8, the labels are too small and some colors chosen look very similar.

8. There are several spelling and grammatical errors. The authors are advised to proofread their manuscript before re-submission.

Reviewer #2: This is a meticuously worked out and written manuscript. It is very well written and quite detailed systematic review on prosepctive trials investigating treatments of radiation dermatitis. Conclusions are clear.

There is only thing, I would ask to adapt, change: there are two references in Brazilian, references number 13 and 14. Authors give non-English publications as an exclusion criterion. They should therefore be removed or substituted by other references. There are good English publications on the use of Calendula.

6. PLOS authors have the option to publish the peer review history of their article (what does this mean?). If published, this will include your full peer review and any attached files.

Reviewer #1: No

Reviewer #2: No

---

## [Author Response · Author response to Decision Letter 0]

11 Dec 2023

December 9, 2023 

Dr. Jingxian Ding

Academic Editor of PLOS ONE

Revision manuscript-" Comparison of the efficacy among different interventions for radiodermatitis: A Bayesian network meta-analysis of randomized controlled trials (Submission ID PONE-D-23-26099)" 

Dear Dr. Ding:

 Thanks for you and the reviewers’ comments of our manuscript titled as Comparison of the efficacy among different interventions for radiodermatitis: A Bayesian network meta-analysis of randomized controlled trials. 

 Your professional advices are helpful for us to improve the quality of the paper. We sincerely accepted and appreciated the reviewer’ comments. We have answered the reviewer’ questions point by point and revised our paper correspondingly. These changes will not influence the content and framework of the paper. And here we did not list the changes but marked in yellow in the revised manuscript. We appreciate for Editors/Reviewers’ warm work earnestly and hope that the correction will meet with approval.

 We look forward to hearing from you again.

Sincerely,

Ying Guan, MD, PhD.

Corresponding Author

Mailing address: No 71, Hedi Road, Nanning, Guangxi 530021, P.R. China, Department of Radiation Oncology, Guangxi Medical University Cancer Hospital 

Email: 386927552@qq.com

Response to the comments of reviewer #1

This study is a Bayesian network meta-analysis of various interventions for the management of radiodermatitis. Despite being the most common side-effect of radiotherapy, the optimal management strategy for radiodermatitis remains elusive. The authors have used a network meta-analysis (NMA) approach to rank different treatment strategies in terms of both clinician and patient reported outcomes. The use of NMA is appropriate in this scenario and the study was conducted in accordance with the established guidelines for conducting systematic reviews and meta-analyses. The results of this study will offer clinicians much-needed information to make evidence-based decisions in managing radiodermatitis. However, the authors need to address the following issues:

1. The authors make conflicting statements about the presence of publication bias in the quality assessment and results sections.

Reply:

We are very sorry for this error and have corrected it in the revised manuscript and marked them in yellow.

2. Under study selection and data collection, the authors report that Any disagreements were resolved through “team discussions”. More clarification should be provided on this. 

Reply:

Thanks for your careful reading our paper. A team discussion means that all four authors/researchers (YG, SL, ACL, and WQC) participated in the discussions. This sentence “or consultation with a third researcher, ACL" is repetitive and may cause misunderstanding, so we delete it. And an explanation of the team discussion is supplemented in parentheses immediately following. We have also made corresponding changes in the revised manuscript and marked them in yellow.

3. In their literature search, the authors did not use other common terms for RD such as skin reactions or skin burns. This could result in failure to retrieve some eligible studies.

Reply:

Thanks a lot for comments. The medical subject headings (Mesh)，Emtree and entry terms were used as search strategies. The complete search strategy and the results can be found in S1 Appendix. The outcomes would be improved if they were coupled with data from other common terms for RD such as skin reactions or skin burns in literature search. But it should be highlighted that Mesh and entry terms were used as search the most popular strategies.

4. The authors can improve the clarity and readability of the manuscript by addressing the following issues:

• The pathogenesis of RD should be written in a less speculative way with reference to specific dermal components that are damaged by IR as well as involved signaling molecules. Also, authors should cite their sources here.

• The discussion is too long and difficult for the reader to follow. The authors should make it more concise and limit the discussion to results of their study.

• The description of NMA in the background should be condensed to one paragraph. Also, the authors should provide a preview of the risk factors for RD can provide a better context to the problem.

• There are several instances where the same information is repeated in the background, methodology, and discussion. The authors should rectify this.

Reply:

Thanks very much for the reviewer's suggestion. We have made corresponding changes in the revised manuscript and marked them in yellow.

(1) We have appropriately modified the background (introduction) section on the pathogenesis of RD to make it more understandable.

(2) In order to clarify the discussion of the 14 RD interventions, we tried our best to make appropriate modifications according to the reviewer's comments, deleting some repeated paragraphs and simplifying sentences.

(3) We condense the NAM's description to one paragraph. Also, we provide a preview of the risk factors for RD.

(4）We deleted the same information is repeated in the background (introduction) section, methodology, and discussion. 

5. The authors should ensure appropriate sources are cited. For instance, the opening statements in the background are not supported by any citations.

Reply:

Thanks for your suggestion. We do check the relative reference again and ensure appropriate sources are cited through the full text. We have made corrections in revised manuscript and have also marked them in yellow.

6. In tables 2B-2E, the authors report too many decimal places in the mean differences and 95% credible intervals.

Reply:

Thanks for your suggestion. We do agree with you and round the data in Table 2B-2E, keeping two decimal places.

7. In Figure 8, the labels are too small and some colors chosen look very similar.

Reply:

Thank reviewer for this comment. We are sorry for the above-mentioned image defects due to software limitations and over-compression of image size factors. We modified it by splitting Figure 8 into individual files (A-E) to make it more legible and readable. We have re-uploaded a clearer version of Fig 8A-8E.

8. There are several spelling and grammatical errors. The authors are advised to proofread their manuscript before re-submission.

Reply:

Thank you for your valuable and thoughtful comments. We have carefully checked and improved the English writing in the revised manuscript. 

Response to the comments of reviewer #2

1. This is a meticuously worked out and written manuscript. It is very well written and quite detailed systematic review on prosepctive trials investigating treatments of radiation dermatitis. Conclusions are clear.

There is only thing, I would ask to adapt, change: there are two references in Brazilian, references number 13 and 14. Authors give non-English publications as an exclusion criterion. They should therefore be removed or substituted by other references. There are good English publications on the use of Calendula.

Reply:

Thanks a lot for comments. We agree with reviewer’s suggestion and have made modifications in References section as requested.

[13] Pommier P, Gomez F, Sunyach MP, D'Hombres A, Carrie C, Montbarbon X. Phase III randomized trial of Calendula officinalis compared with trolamine for the prevention of acute dermatitis during irradiation for breast cancer. J Clin Oncol. 2004 Apr 15;22(8):1447-53. doi: 10.1200/JCO.2004.07.063. PMID: 15084618.

[14] Kumar S, Juresic E, Barton M, Shafiq J. Management of skin toxicity during radiation therapy: a review of the evidence. J Med Imaging Radiat Oncol. 2010 Jun;54(3):264-79. doi: 10.1111/j.1754-9485.2010.02170.x. PMID: 20598015.

Response to the editorial comments

A rebuttal letter that responds to each point raised by the academic editor and reviewer(s). You should upload this letter as a separate file labeled 'Response to Reviewers'. 

A marked-up copy of your manuscript that highlights changes made to the original version. You should upload this as a separate file labeled 'Revised Manuscript with Track Changes'.

An unmarked version of your revised paper without tracked changes. You should upload this as a separate file labeled 'Manuscript'.

Reply:

Yes, we do revise our paper in response to the reviewer' and editor’s comments point-by-point, containing both the original query followed by our replies. And we also modified manuscript accordingly. We have uploaded response file as a 'Point-by-point response' file. All changes to the manuscript were highlighted or indicated by using tracked changes. The electric files of the revised manuscript with track changes as well as figures, respectively after we completed revisions.

2. Reply:

The financial disclosure has not been changed. We resubmit the Fig 8 in accordance with the requirements of the submission format of figure files.

3. If applicable, we recommend that you deposit your laboratory protocols in protocols.io to enhance the reproducibility of your results. Protocols.io assigns your protocol its own identifier (DOI) so that it can be cited independently in the future. For instructions see: https://journals.plos.org/plosone/s/submissionguidelines#loc-laboratory-protocols. Additionally, PLOS ONE offers an option for publishing peer-reviewed Lab Protocol articles, which describe protocols hosted on protocols.io.

Reply:

 Thank you for your recommendation. The current study protocol was registered in PROSPERO with the registration number CRD42023428598.

Reply:

We've read PLOS ONE formatting sample PDF file about title, authors, affiliations, main body, and file naming. The manuscript has also been modified in accordance with its relevant requirements. A marked-up copy of manuscript that yellow-highlight changes made to the original version. We also upload this as a separate file labeled 'Revised Manuscript with Track Changes'.

Reply:

Thanks for your suggestion. All data generated or analyzed during this study are included in this published article (and its Supporting Information files).

6. PLOS requires an ORCID iD for the corresponding author in Editorial Manager on papers submitted after December 6th, 2016. Please ensure that you have an ORCID iD and that it is validated in Editorial Manager. To do this, go to ‘Update my Information’ (in the upper left-hand corner of the main menu), and click on the Fetch/Validate link next to the ORCID field. This will take you to the ORCID site and allow you to create a new iD or authenticate a preexisting iD in Editorial Manager.

Reply:

Yes, I do have an ORCID iD. I have followed the instructions to fetch/validate my ORCID iD by updating my information.

7. Please include captions for your Supporting Information files at the end of your manuscript, and update any in-text citations to match accordingly.

Reply:

 Yes, we include captions for our Supporting Information files at the end of our revised manuscript, and update any in-text citations to match accordingly.

Reply:

We re-checked the list of references and no documents were retracted. At the same time, the reference format was modified according to the requirements of PLOS ONE journal. Changes to the list of references are mentioned in the rebuttal letter accompanying the revised manuscript.

---

## [Decision Letter · Decision Letter 1]

22 Jan 2024

Comparison of the efficacy among different interventions for radiodermatitis: A Bayesian network meta‑analysis of randomized controlled trials

PONE-D-23-26099R1

Dear Dr. Guan,

We’re pleased to inform you that your manuscript has been judged scientifically suitable for publication and will be formally accepted for publication once it meets all outstanding technical requirements.

Kind regards,

Huijuan Cao, Ph.D.

Academic Editor

PLOS ONE

Additional Editor Comments (optional):

Reviewers' comments:

Reviewer's Responses to Questions

**Comments to the Author**

1. If the authors have adequately addressed your comments raised in a previous round of review and you feel that this manuscript is now acceptable for publication, you may indicate that here to bypass the “Comments to the Author” section, enter your conflict of interest statement in the “Confidential to Editor” section, and submit your "Accept" recommendation.

Reviewer #1: All comments have been addressed

Reviewer #2: All comments have been addressed

2. Is the manuscript technically sound, and do the data support the conclusions?

Reviewer #1: Yes

Reviewer #2: Yes

3. Has the statistical analysis been performed appropriately and rigorously? 

Reviewer #1: Yes

Reviewer #2: I Don't Know

4. Have the authors made all data underlying the findings in their manuscript fully available?

Reviewer #1: Yes

Reviewer #2: Yes

5. Is the manuscript presented in an intelligible fashion and written in standard English?

Reviewer #1: Yes

Reviewer #2: Yes

6. Review Comments to the Author

Reviewer #1: All comments made during the previous round of review have been appropriately addressed. The current version is suitable for publication.

Reviewer #2: All comments of reviewers have been addressed by the authors and changes have been made. Responses are sufficient

7. PLOS authors have the option to publish the peer review history of their article (what does this mean?). If published, this will include your full peer review and any attached files.

Reviewer #1: No

Reviewer #2: No

---

## [Editor Report · Acceptance letter]

30 Jan 2024

PONE-D-23-26099R1 

PLOS ONE

Dear Dr. Guan, 

I'm pleased to inform you that your manuscript has been deemed suitable for publication in PLOS ONE. Congratulations! Your manuscript is now being handed over to our production team.

Kind regards, 

on behalf of

Dr. Huijuan Cao 

Academic Editor

PLOS ONE